# OmegAMP: Targeted AMP Discovery via Biologically Informed Generation

## Abstract

Deep learning-based antimicrobial peptide (AMP) discovery faces critical challenges such as limited controllability, lack of representations that efficiently model antimicrobial properties, and low experimental hit rates. To address these challenges, we introduce OmegAMP, a framework designed for reliable AMP generation with increased controllability. Its diffusion-based generative model leverages a novel conditioning mechanism to achieve fine-grained control over desired physicochemical properties and to direct generation towards specific activity profiles, including species-specific effectiveness. This is further enhanced by a biologically informed encoding space that significantly improves overall generative performance. Complementing these generative capabilities, OmegAMP leverages a novel synthetic data augmentation strategy to train classifiers for AMP filtering, drastically reducing false positive rates and thereby increasing the likelihood of experimental success. Our *in silico* experiments demonstrate that OmegAMP delivers state-of-the-art performance across key stages of the AMP discovery pipeline, enabling us to achieve an unprecedented success rate in wet lab experiments. We tested 25 candidate peptides, 24 of them (96%) demonstrated antimicrobial activity, proving effective even against multi-drug resistant strains. Our findings underscore OmegAMP's potential to significantly advance computational frameworks in the fight against antimicrobial resistance.

## 1 Introduction

Antimicrobial resistance, ranking as the third leading cause of death in 2019 (Murray et al., 2022), poses a critical threat to human health. As existing therapeutics prove insufficient, antimicrobial peptides (AMPs) emerge as a promising alternative with transformative potential. These short, biologically active sequences offer broad-spectrum antimicrobial activity, with a reduced likelihood of resistance development compared to traditional antibiotics (Fjell et al., 2012). Their function is governed by key physicochemical properties—such as *charge*, *length*, and *hydrophobicity*—which influence peptide structure, membrane interaction, and ultimately, antimicrobial efficacy. Controlling these features is essential for maximizing antimicrobial efficacy and ensuring synthesizability.

Discovering AMPs is resource-intensive, making computational methods essential to save time and minimize costs. These computational approaches generally fall into two categories: generative models for *de novo* AMP design (Szymczak et al., 2023; Chen et al., 2024; Van Oort et al., 2021), and discriminative models that distinguish peptides with respect to antimicrobial activity (Li et al., 2022; Lawrence et al., 2021; Veltri et al., 2018). Despite important progress, current methods face key challenges: *(i)* **Limited controllability:** Existing generative models lack mechanisms for inherent, nuanced control, preventing them from directly generating AMPs with desired physicochemical and functional properties. This, in turn, constrains the exploration of diverse activity profiles and the generation of distinct peptides, such as target-specific AMPs (Szymczak & Szczurek, 2023). *(ii)* **Ineffective embedding:** An adequate peptide embedding is crucial for generative fidelity and fine-grained control. Yet, existing approaches present limitations: while biologically agnostic embeddings (e.g., one-hot) lack useful inductive biases, complex latent spaces of protein language models hinder precise conditioning and control. Consequently, neither effectively incorporates biological information to facilitate the targeted generation of AMPs. *(iii)* **Low Experimental Hit Rates:** Effective peptide selection is crucial for reducing the cost of discovering novel therapeutics, yet current frameworks often suffer from low success rates (Wan et al., 2024). This is primarily due to ineffective generation and unreliable classifier-based filtering that lack specificity, struggle with

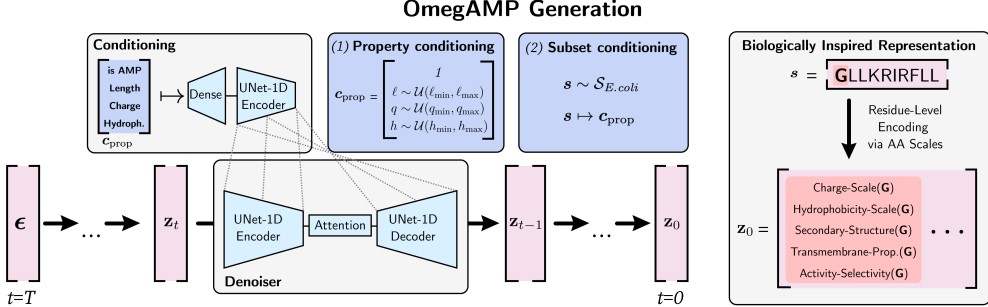

Figure 1: OmegAMP provides practitioners the ability to generate AMPs conditioned on key physicochemical properties, like length, charge, and hydrophobicity. Our generative model enables more complex objective targeting via Property and Subset conditioning.

shuffled sequences, produce many false positives, and overfit to limited training data (Porto et al., 2022).

Addressing these limitations, we introduce OmegAMP, a framework that synergistically integrates controlled AMP generation, see Fig. 1, with robust filtering to reliably design candidate AMPs. Our contributions can be summarized as follows:

1. **A versatile conditioning mechanism that enables the fine-grained control over desired AMP attributes**. We showcase a first-in-class capability in generating AMPs with an increased probability of targeting specific bacterial species, highlighting the potential for targeted therapeutics.

2. **A novel biologically-inspired peptide embedding scheme**, leveraged by our diffusion model for state-of-the-art performance in de novo AMP generation.

3. **An unprecedented experimental hit rate achieved through controllable generation and stringent filtering.** We complement our proposed generative process with a synthetic data augmentation strategy to train classifiers that drastically reduce false positive rates. Our approach bridges the gap between *in silico* design and wet-lab validation by increasing the likelihood of experimental success.

## 2 RELATED WORK

**AMP Generation** Several machine learning approaches have been developed for generating novel AMP sequences, differing in their underlying models, data representations, and ability to incorporate desired properties (conditional generation). Initial methods used Variational Autoencoders (VAEs), e.g. HydrAMP (Szymczak et al., 2023) employed a conditional VAE (cVAE) with specialized regularization, conditioning generation on predicted AMP activity. AMPGAN (Van Oort et al., 2021) combined VAEs with Generative Adversarial Networks (GANs) to enable conditioning based on factors like target microbes and MIC values. More recently, diffusion models have been applied. AMP-Diffusion, Diff-AMP, and ProT-Diff (Chen et al., 2024; Wang et al., 2024a;b) perform diffusion directly within the latent spaces derived from large protein language models (e.g., ESM2 (Lin et al., 2022), ProtT5 (Elnaggar et al., 2021)), though typically without conditional control.

**AMP Classification** Classification methods can be broadly categorized as either deep learning-based or ensemble tree-based, using features like position-specific encodings (e.g., one-hot, language model embeddings), global physicochemical properties, and sequence statistics. Early deep learning approaches include AMPScanner (Veltri et al., 2018), using RNNs and CNNs on numerical encodings, AMPlify (Li et al., 2022), employing BiLSTMs on one-hot vectors, and AMPpredMFA (Li et al., 2023), combining LSTMs and Attention layers to operate at the residue and di-residue level. Concurrently, ensemble methods like amPEPpy (Lawrence et al., 2021) proved effective, using Random Forests with global sequence descriptors that combine physicochemical and sequence-based features. Recently, methods like SenseXAMP (Zhang et al., 2023) have used neural networks to fuse language model embeddings and protein descriptors to perform classification. Specialized classifiers that prioritize false-positive rates have also been developed. PyAMPA (Ramos-Llorens et al., 2024), for example, operates on tokenized di-peptide features with a biologically motivated filtering.

## 3 OMEGAMP

### 3.1 GENERATION

Our generative model, denoted as $\mathcal{M}_\theta$, is a Denoising Diffusion model which incorporates two main innovations: a novel biologically informed **peptide embedding** and a flexible **conditioning scheme** designed for diverse target objectives. For stable and diverse generation, particularly under conditional constraints, $\mathcal{M}_\theta$ incorporates self-conditioning (Chen et al., 2022) and the CADS sampler (Sadat et al., 2023). We provide a detailed background on diffusion and our general conditioning principles in App. A.1.

At its core, the denoising network is a 1D UNet (Ronneberger et al., 2015) operating on noisy instances of our biologically informed peptide embedding. This UNet architecture integrates linear attention layers (Katharopoulos et al., 2020) and a central self-attention mechanism (Vaswani et al., 2023), drawing inspiration from the TransUNet architecture (Chen et al., 2021). Within this, we integrate a flexible conditioning scheme by injecting conditioning information throughout all network layers. Following the approach of Rombach et al. (2022), representations of the conditioning vector are concatenated not only with the initial noisy input $\mathbf{z}_t$ but also with intermediate feature maps at various stages of the denoising network, as illustrated in Fig. 1. This methodology enables the model to directly incorporate relevant target peptide properties into the denoising process.

Comprehensive details regarding model hyperparameters and the datasets used for training are available in App. B.1 and C, respectively.

**Peptide Embedding** To provide a biologically informed space for the generative model to operate on, we propose an embedding scheme, where each amino acid $a \in \mathcal{A}$ (along with a special padding token PAD) is mapped to a $K$-dimensional vector $E(a) = [f_1(a), f_2(a), \ldots, f_K(a)]$. This embedding is constructed by applying a set of $K$ pre-selected physicochemically-inspired scales $\{f_1, f_2, \ldots, f_K\}$, where each scale $f_i : \mathcal{A} \cup \{\text{PAD}\} \to \mathbb{R}$ transforms an amino acid into a real-valued biochemical property relevant to antimicrobial function. Conversely, to decode the embeddings back to amino-acids, the amino acid encoding $E(a)$ must be invertible, a property ensured if $E(a)$ is injective. The decoding procedure then operates by identifying the amino acid $a'$ whose $K$-dimensional encoding $E(a')$ is $L_2$-closest to the target embedding vector. A concrete implementation for the encoding and decoding procedure is provided in App. B.2.

*Selection of Biologically Informative Scales* The choice of the specific amino acid scales for our embedding $E(a)$ is crucial for capturing properties directly relevant to antimicrobial activity and peptide structure. We therefore curated a set of 5 scales designed to provide complementary biochemical information pertinent to AMP function: *(1)* **Hydrophobicity:** The Wimley-White scale (Wimley & White, 1996) quantifies amino acid membrane affinity and insertion propensity. *(2)* **Charge:** The isoelectric point (pI) indicates the pH at which an amino acid carries no net electrical charge, helping to distinguish acidic and basic residues. *(3-4)* **Structural Propensities:** The Levitt scale (Levitt, 1978) captures secondary structure tendencies, while the Transmembrane Propensity scale (Zhao & London, 2006) informs about transmembrane helix formation. *(5)* **Antimicrobial Correlation:** The Average Amino Acid Surface Area Index (AASI) (Juretic et al., 2009) is directly correlated with antimicrobial activity and selectivity. Together, these scales imbue our embedding $E(a)$ with a multifaceted biochemical profile essential for effective AMP generation; see App. E for a background on amino acid scales.

**Conditioning Scheme** Effective AMP design requires the control of activity, physiocochemical and structural features. We propose a conditioning scheme that aligns well with our embedding and enables targeted generation through explicit control of underlying physicochemical attributes. For any given peptide sequence $\boldsymbol{s}$, we define a function $\text{cond}(\boldsymbol{s})$ that maps the input sequence to a vector representing key controllable properties: being an AMP, length, net charge and overall hydrophobicity.

$$\text{cond}(\boldsymbol{s}) := \begin{pmatrix} \mathbf{1}_{\text{AMP}}(\boldsymbol{s}) \\ |\boldsymbol{s}| \\ \text{Charge}(\boldsymbol{s}) \\ \text{Hydroph.}(\boldsymbol{s}) \end{pmatrix} \quad . \tag{1}$$

Importantly, $\text{cond}(\boldsymbol{s})$ can be readily obtained, as the AMP property is set to 1 if the peptide originates from curated AMP databases, and is set to 0 otherwise. The length, charge and hydrophobicity

properties can be obtained via known biological algorithms. Our focus on deterministically accessible properties prevents the risk of conditioning on noisy labels, a common pitfall when using trained classifiers on scarce data (Szymczak & Szczurek, 2023). During inference, a user can specify a *target profile* by providing desired values for these properties (or indicating an omission $\varnothing$ for unconstrained attributes), which then forms the conditional input to the generative model, denoted by $\boldsymbol{c}$. Importantly, this framework retains generality, as AMP unconditional generation is achieved by omitting all properties but the AMP property, which is set to 1.

To make our generative model $\mathcal{M}_\theta$ adhere to such target profiles, we introduce a conditional training objective. This objective allows the model to learn the relationship between peptide sequences and varying subsets of their properties. To do this, for each training sequence $\boldsymbol{s}$, we first compute the full property vector $\text{cond}(\boldsymbol{s})$ and then generate multiple conditioning vectors $\boldsymbol{c}$ by selectively masking its elements in various ways. Let $m$ be a binary mask sampled from a distribution $\mathcal{D}_{\text{mask}}$. The modified conditioning vector is then $\boldsymbol{c} = \text{cond}(\boldsymbol{s}) \diamond m$, where $\diamond$ symbolizes an operation that applies the mask (e.g., replacing masked elements with a special token). The general conditional loss is then:

$$\mathcal{L}_{\text{conditional}}(\theta) \coloneqq \mathbb{E}_{\boldsymbol{s}\sim Q, m\sim\mathcal{D}_{\text{mask}}} \left[ \mathcal{L}_{\text{instance}}(\theta, \boldsymbol{s}, \text{cond}(\boldsymbol{s}) \diamond m)) \right], \tag{2}$$

where $\mathcal{L}_{\text{instance}}(\theta, \boldsymbol{s}, \boldsymbol{c})$ is an instance-level loss quantifying the error for a single data sample $\boldsymbol{s}$, sampled from a distribution of sequences $Q$, given conditioning information $\boldsymbol{c}$, specific to the generative framework. Let $\boldsymbol{E}(\boldsymbol{s})$ denote the biologically-inspired embedding of peptide $\boldsymbol{s}$, for diffusion models the instance loss equates to:

$$\mathcal{L}_{\text{instance}}(\theta, \boldsymbol{s}, \mathbf{c}) \coloneqq \mathbb{E}_{t,\mathbf{z}_t} \left[ \left\| \hat{\boldsymbol{E}}_\theta(\mathbf{z}_t, t, \mathbf{c}) - \boldsymbol{E}(\boldsymbol{s}) \right\|_2^2 \right] . \tag{3}$$

In our specific implementation, the AMP property is always included in $\boldsymbol{c}$. For the remaining three properties (Length, Charge, Hydrophobicity), we uniformly at random choose to keep $k \in \{0, 1, 2, 3\}$ of them active, effectively sampling a binary mask $m$ that selects $k$ properties. This training strategy encourages the model to learn how each specified property (and combinations thereof) influences sequence generation, enabling versatile conditional control.

***Generalizability***   Our conditioning scheme facilitates targeted exploration of the AMP landscape. Given that for any AMP target distribution, we can characterize its sample space by the true underlying set of desired sequences $\mathcal{S}_{\text{target}}$, each such target distribution possesses a corresponding set of property vectors $\mathcal{C}_{\text{target}} = \{\text{cond}(\boldsymbol{s}) \mid \boldsymbol{s} \in \mathcal{S}_{\text{target}}\}$. By conditioning on such vectors $\boldsymbol{c} \in \mathcal{C}_{\text{target}}$, and assuming a sufficiently expressive generative model $\mathcal{M}_\theta$, we can generate a candidate set $\mathcal{S}_G$ that is highly enriched with, and ideally contains, the desired sequences ($\mathcal{S}_{\text{target}} \subseteq \mathcal{S}_G$). The subsequent task then becomes effectively filtering $\mathcal{S}_G$, using computational classifiers or experimental validation, to isolate sequences that best match $\mathcal{S}_{\text{target}}$. Thus, the main challenge for enhancing generation for complex targets lies in effectively translating criteria into a representative set of conditioning vectors that guide $\mathcal{M}_\theta$ towards the desired region of the peptide space. These vectors should aim to effectively cover $\mathcal{S}_{\text{target}}$ while ensuring the generated candidate pool is focused enough to reduce reliance on extensive and costly downstream filtering.

***Conditioning Strategy***   The challenge of translating complex criteria characterizing $\mathcal{S}_{\text{target}}$ into a set of conditioning vectors representative for $\mathcal{C}_{\text{target}}$ motivates two practical methodologies for generating appropriate conditioning inputs.

*(1)* **Property Conditioning (PC)** allows practitioners to translate expert knowledge into specified ranges for the properties within $\text{cond}(\cdot)$. Conditioning vectors are then formed by sampling each property value independently from its pre-defined range. This offers flexibility but may not capture inherent correlations between properties.

*(2)* **Subset Conditioning (SC)** enables the use of a known set of example sequences $\mathcal{S}_{\text{sample}} = \{\boldsymbol{s}'_1, \ldots, \boldsymbol{s}'_M\}$ from $\mathcal{S}_{\text{target}}$ (e.g., a set of known peptides that are active against *E. coli*). Here, conditioning vectors are directly computed $\{\text{cond}(\boldsymbol{s}'_1), \ldots, \text{cond}(\boldsymbol{s}'_M)\}$. Sampling conditional inputs from this set implicitly preserves property correlations present in the target sequences, offering an empirical approximation of a portion of $\mathcal{C}_{\text{target}}$.

These approaches enable distinct strategies for targeted generation: either by directly defining desired property ranges (PC) or by leveraging properties of known peptides that satisfy the intended target (SC).

## 3.2 FILTERING

To obtain highly confident and reliable filters, we trained XGBoost classifiers using augmented datasets that contain synthetic negative sequences, complemented with a custom loss function that balances the contributions of sequences with known labels and synthetic sequences. We provide additional details regarding the data used for training, the classifier features, and hyperparameters in App. C, D.1 and D.2, respectively. The trained classifiers include a general classifier for AMP property, as well as strain- and species-specific classifiers that predict activity against specific bacterial targets. In App. F, we expand on the definitions of strain and species activity.

**Synthetic Negatives For Classifier Training**  To enhance the robustness of AMP classifiers and minimize false positives, we incorporate synthetic data that, by construction, are non-AMPs. Let $\mathcal{S}_L^1 = \{s \in \mathcal{A}^L \mid y = 1\}$ represent the set of AMP sequences of length $L$. We propose three mechanisms of constructing synthetic negatives:

*(1)* **Purely Random Sequences (R):** Generate sequences $s = (a_1, \ldots, a_L)$, where each amino acid $a_i \sim U(\mathcal{A})$ is independently drawn from the uniform distribution over the amino acid space $\mathcal{A}$.

*(2)* **Shuffled AMP Sequences (S):** Given a known AMP sequence $s \in \mathcal{S}_L^1$, generate a shuffled sequence $\pi(s)$, where $\pi \in \mathcal{P}_L$ is a random permutation from the symmetric group $\mathcal{P}_L$.

*(3)* **Mutated AMP Sequences (M):** Starting from a known AMP sequence $s \in \mathcal{S}_L^1$, randomly select 5 distinct positions $\mathbf{p} = \{p_1, p_2, \ldots, p_5\}$, with $1 \leq p_k \leq L$ for $k = 1, \ldots, 5$. For each selected position $p_k \in \mathbf{p}$, replace the amino acid $a_{p_k}$ with a new amino acid $a'_{p_k} \sim U(\mathcal{A} \setminus \{a_{p_k}\})$.

Random sequences maintain the original length profile while disrupting functional patterns, preventing the model from relying solely on length distributions. Shuffled sequences preserve permutation-invariant properties such as overall charge and hydrophobicity but modify the sequence order, prompting the model to look beyond these global characteristics. Mutated sequences introduce controlled changes at specific positions while largely preserving the original sequence context, discouraging over-reliance on individual residues. In App. G, we provide theoretical and empirical motivations for the expected inactivity of those sequences.

Integrating experimentally validated peptides (Experimentally Validated, EV) with synthetic sequences possessing inferred labels is challenging given the small but non-zero probability of mislabelling in the synthetic data. To address this varying data quality alongside class imbalance, we propose a *weighted binary cross-entropy loss*, see App. D.3, that weighs EV sequences more favorably than non-EV sequences. Therefore, allowing the use of an expanded version of our limited labeled set while prioritizing the accurate distinction of experimentally verified peptides.

**Challenging Negative Controls for Evaluation**  Due to the limited number of EV inactive sequences ($< 1000$ sequences) and to emulate the real-world imbalancedness between AMP/Non-AMP sequences, we select three additional sources of inactive sequences for model evaluation:

*(1)* **Signal Peptides (Sig)**: Biologically functional peptides that guide proteins to their proper cellular destinations for secretion or transport.

*(2)* **Metabolic Peptides (Met)**: A class of functional peptides that regulate metabolic pathways and maintain the body's energy homeostasis.

*(3)* **Added-Deleted AMP Sequences (AD):** Starting from a known AMP sequence $s \in \mathcal{S}_L^1$, sequentially apply 5 modifications, each randomly chosen to be an insertion of an amino acid $a' \sim U(\mathcal{A})$ at a random position, or a deletion of an existing amino acid at a random position.

Signal and metabolic peptides present a challenge for models trained to classify AMPs, as they require the model to differentiate between antimicrobial, signal, and metabolic functions. These types of negative sequences are never seen during the training of AMP classifiers, making them a fair test of performance. Even more difficult to detect are Added-Deleted sequences, which mimic imperfect, yet AMP-like, outputs from generative models. Unlike the other synthetic Non-AMP sources, AD sequences are not included in the classifiers' training data. They retain the core AMP structure but contain subtle changes that render them inactive, making them a particularly difficult test for the model's ability to discern true functionality.

Table 1: Performance on a held-out test set of EV AMPs and various non-AMP sources. Robustness is the misclassification rate (%) on these challenging negatives. Grey columns denote test sets from sources entirely unseen during training (Sig, Met, AD), while other synthetics (R, S, M) are unseen sequences from Non-AMP sources known to the model.

| | **Performance Metrics** | | | | | **Robustness (↓) (Misclassification Rate %)** | | | | | |
| | | | | | | **Bio Non-AMPs** | | **Synthetic Non-AMPs** | | | |
| **Model** | **AUPRC (↑)** | **Prec@100 (↑)** | **TPR (↑)** | **FPR (↓)** | **LR+ (↑)** | **Sig** | **Met** | **AD** | **R** | **S** | **M** |
|---|---|---|---|---|---|---|---|---|---|---|---|
| amPEPpy | 14.8 | 50.6 | 94.2 | 40.1 | 2.4 | 15.1 | 38.0 | 82.8 | 31.2 | 87.6 | 73.7 |
| AMPlify | 16.7 | 41.6 | 96.1 | 36.0 | 2.7 | 7.2 | 31.8 | 87.3 | 23.3 | 86.4 | 80.9 |
| AMPpredMFA | 5.5 | 6.8 | **99.4** | 77.5 | 1.3 | 65.2 | 75.7 | 99.3 | 97.7 | 99.7 | 99.5 |
| AMPScanner | 9.7 | 12.7 | 96.3 | 51.3 | 1.9 | 37.6 | 41.1 | 81.5 | 25.8 | 81.5 | 80.4 |
| HydrAMP-AMP | 11.9 | 16.0 | 94.9 | 53.7 | 1.8 | 37.7 | 30.7 | 84.2 | 33.7 | 86.1 | 75.0 |
| HydrAMP-MIC | 14.9 | 15.2 | 81.6 | 21.3 | 3.8 | 8.9 | 2.3 | 46.0 | 2.5 | 56.8 | 43.2 |
| SenseXAMP-classifier | 14.8 | 20.0 | 97.9 | 67.5 | 1.5 | 49.1 | 55.2 | 90.1 | 48.4 | 88.3 | 90.4 |
| PyAMPA | 4.2 | 13.4 | 27.0 | 13.4 | 2.0 | 6.7 | 6.9 | 31.2 | 3.8 | 27.9 | 20.4 |
| **OmegAMP** | **56.9** | **90.4** | 43.5 | **0.3** | **138.1** | **0.0** | **0.4** | **0.5** | **0.0** | **0.4** | **0.7** |

# 4 EXPERIMENTS

## 4.1 AMP CLASSIFICATION

In this subsection, we analyze OmegAMP classifier's ability to recognize antimicrobial activity in the context of AMP/non-AMP distinction. To view the capabilities of OmegAMP classifiers in specialized settings like species- and strain-specific activity, please refer to App. J.1.

**False positive rates are drastically reduced across natural and synthetic non-AMP sources** To evaluate our proposed classification scheme from Sec. 3.2, we compare OmegAMP with existing baseline models, using the model checkpoints released alongside their original papers: amPEPpy, AMPlify, AMPpredMFA, AMPScanner, two classifier models from HydrAMP, SenseXAMP-classifier, and PyAMPA (Lawrence et al., 2021; Zhang et al., 2023; Li et al., 2023; Veltri et al., 2018; Szymczak et al., 2023; Li et al., 2022; Ramos-Llorens et al., 2024). In order to assess the classifiers' robustness to challenging false positives, we train the general OmegAMP classifier on a union of EV data and negative synthetic datasets (see App. C). The training is conducted using 5-fold cross-validation, with 20% of the EV data held out for testing, and all results are averaged across the 5 folds.

For robustness, we focus on the models' ability to identify AMPs within sets of challenging non-AMPs. The evaluation dataset is composed of EV data and challenging negative sequences including ∼10k Signal and ∼17k Metabolic peptides extracted from Peptipedia (Cabas-Mora et al., 2024), as well as 10k AD synthetic sequences. This selection was performed to make our evaluation reflect real-world requirements. We report the false positive rate (FPR), true positive rate (TPR), Precision@100 (precision for the top 100 sequences by model logits), and the Positive Likelihood ratio (LR+ = TPR/FPR), a metric widely used to assess the practical relevance of diagnostic tools (Deeks & Altman, 2004). We provide the Area Under the Precision-Recall Curve (AUPRC) as this metric is independent of threshold selection. For completeness, we additionally report the fraction of sequences predicted to be an AMP for each source of non-AMPs, including the Random, Shuffled and Mutated Sequences.

Tab. 1 shows that while baseline classifiers achieve high TPRs, they tend to misclassify non-AMPs as AMPs, leading to large FPRs and low LR+ ratios. This weakness is particularly exposed when evaluating the robustness of these methods to challenging natural and synthetic Non-AMP sequences. In these settings baseline methods tend to consistently misclassify non-AMPs as AMPs, whereas, OmegAMP has misclassification rates $< 1\%$ for all considered non-AMP sources. Additionally, the superior Prec@100 (90.4%) and AUPRC (56.9) scores are practically significant, as they ensure that the highest-scoring sequences are highly likely to be true AMPs. Overall, our findings establish OmegAMP as a reliable filter of inactive peptides, therefore, meeting the requirements of real-world discovery pipelines.

## 4.2 AMP GENERATION

We next present a detailed analysis of OmegAMP's capabilities to generate realistic AMP sequences and a thorough inspection of its ability to meet pre-specified physicochemical criteria.

Table 2: Performance comparison across generative models. We report the percentage of predicted positives (HydrAMP-MIC classifier, OmegAMP classifier), along with Fitness Score, Diversity, Uniqueness and Novelty. Results signaled with * come from $k$-fold-cross validation averaging.

| Gen. Model | HydrAMP-MIC | OmegAMP Class. | Fitness Score | Diversity | Uniqueness | Novelty |
|---|---|---|---|---|---|---|
| EV AMPs (*Data*) | 81.6 | 43.5* | 0.16 | 0.62 | - | - |
| AMPGAN | 31.6 | 0.3 | 0.10 | 0.57 | **100** | **100** |
| Diff-AMP | 27.8 | 0.0 | 0.08 | 0.63 | **100** | **100** |
| HydrAMP | 44.1 | 0.0 | 0.09 | **0.70** | **100** | **100** |
| AMP-Diffusion | 42.8 | 2.2 | 0.11 | 0.64 | 91 | **100** |
| OmegAMP | 33.8 | 10.5 | 0.13 | 0.64 | 94 | 98 |
| OmegAMP-PC | **70.2** | 14.8 | **0.16** | 0.60 | 98 | 99 |
| OmegAMP-SC | 64.1 | **16.4** | 0.15 | 0.61 | 95 | 97 |

Table 3: Embedding scheme ablation with respect to generative performance on unconditional metrics. Columns are as in Tab. 2.

| Gen. Model | Omeg. Class. | Fit. Score | Div. | Uniq. |
|---|---|---|---|---|
| Omeg. w/ Numeric | 2.5 | 0.11 | 0.61 | **98** |
| Omeg. w/ One-hot | 6.6 | 0.12 | 0.63 | 97 |
| Omeg. − {charge scale} | 8.4 | 0.12 | **0.64** | 96 |
| Omeg. − {hydroph. scale} | 9.6 | **0.13** | **0.64** | 95 |
| OmegAMP | **10.5** | **0.13** | **0.64** | 94 |

Table 4: Embedding scheme ablation with respect to generative performance on conditional metrics. Metrics consist of MAEs in std units.

| Gen. Model | MAE (in std units) (↓) | | |
|---|---|---|---|
| | Length | Charge | Hydroph. |
| Omeg. w/ Numeric | **0.00** | 0.77 | 0.63 |
| Omeg. w/ One-hot | 0.08 | 0.17 | 0.19 |
| Omeg. − {charge scale} | 0.08 | 0.24 | 0.19 |
| Omeg. − {hydroph. scale} | 0.35 | 0.18 | 0.22 |
| OmegAMP | 0.04 | **0.16** | **0.18** |

**OmegAMP generator achieves state-of-the-art performance**   To contextualize OmegAMP's generative model, see Sec. 3.1, and demonstrate its performance relative to existing AMP generators, we analyse both the antimicrobial activity and sequence diversity of OmegAMP-generated sequences relative to those produced by baselines: AMPGAN, Diff-AMP, HydrAMP, and AMP-Diffusion (Van Oort et al., 2021; Wang et al., 2024a; Szymczak et al., 2023; Chen et al., 2024). We evaluate 50k samples per model when available. In addition, we evaluate OmegAMP in two conditional settings: Property Conditioning, referred to as OmegAMP-PC, with expert-defined property intervals of charge 2 to 10, hydrophobicity −0.5 to 0.8, and lengths between 5 and 30; and Subset Conditioning, labeled as OmegAMP-SC, which incorporates subset conditioning on the EV General AMP sequences described in App. C. To assess these samples, we use metrics for generation quality, diversity, uniqueness and novelty, presented in detail in App. H. Furthermore, we report the antimicrobial potential according to the two best-performing classifiers from Tab. 1, namely HydrAMP-MIC and OmegAMP's classifier. Finally, we provide fitness scores (Li et al., 2024) as a proxy for amphiphacity, a key feature related to AMP activity.

Our comparison in Tab. 2 highlights OmegAMP's superior ability to generate high-quality AMP sequences, as reflected in its higher AMP classification and fitness scores. The low fitness ($\leq 0.11$) exhibited by all baseline models indicates limited functional relevance of their outputs. Moreover, OmegAMP variants with conditional guidance, OmegAMP-PC and OmegAMP-SC, achieve the highest scores as per classifiers HydrAMP-MIC and OmegAMP, as well as the highest fitness, with OmegAMP-PC matching the fitness scores observed for EV AMPs. This suggests that our embedding scheme and conditioning guidance strategies increase the biological quality of generated sequences. Therefore, while all models produce diverse, unique, and novel sequences (diversity: $0.60 \geq$, uniqueness: $\geq 94\%$, novelty: $\geq 97\%$), only OmegAMP combines these characteristics with superior predicted antimicrobial activity, highlighting its effectiveness in the generation of novel AMPs.

**OmegAMP's embedding scheme is crucial for generative performance**   To assess the individual contribution of our biologically-informed peptide embedding from Sec. 3.1, we ablate distinct embedding schemes within OmegAMP, evaluating their impact on unconditional generation quality, identical to Sec. 4.2, and conditional control over key physicochemical properties. These ablated representations include biology-agnostic schemes (Numeric and One-hot encoding) and partial versions of our proposed embedding (removing either charge or hydrophobicity scales). To evaluate the impact on conditional control, we conduct an additional study where models are conditioned on

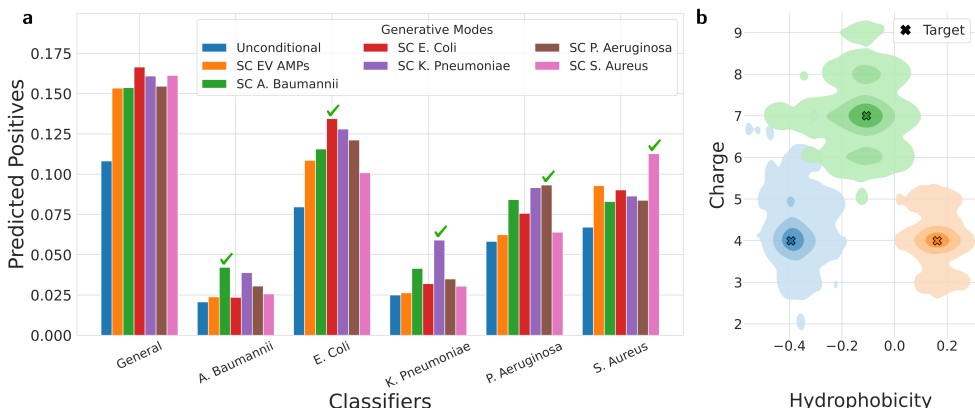

Figure 2: *a)* Subset conditioning shows that generating sequences based on those active against a specific species increases the likelihood of producing active sequences. *b)* Property conditioning reliably generates peptides with charge and hydrophobicity values that approximate the pre-specified target.

only one property at a time (omitting others). For this, we generate 10K samples using conditioning vectors uniformly sampled from the AMP training set. We then compute the Mean Absolute Error (MAE) between the pre-specified conditioning values and the properties of the generated sequences, normalizing each property's MAE by its standard deviation in the training set.

The findings from this ablation study clearly illustrate the benefits of our chosen embedding. Our ablations concerning unconditional generation quality in Tab. 3 demonstrate that replacing our biology-informed embedding with numeric or one-hot encodings substantially reduces AMP prediction rates, as well as fitness scores, and diversity. Furthermore, removing either the charge or hydrophobicity scales from the embedding also results in lowered AMP prediction rates, confirming the criticality of these features for generating high-quality, AMP-like sequences. In terms of conditional control, see Tab. 4, the full OmegAMP model achieves the lowest MAE for charge ($0.16$) and hydrophobicity ($0.18$) and competitive performance for length ($0.04$). The ablations reveal that removing specific scales weakens control over the corresponding property: for instance, individually omitting the charge and hydrophobicity scale increase the MAE from $0.16$ to $0.24$ and $0.18$ to $0.22$, respectively. These results underscore the direct relevance of each scale in enabling precise conditional control over its corresponding property. This property is also preserved in the more challenging multi-conditioning setting, see App. I.4.

We therefore conclude that our biologically-informed embedding is an essential component underpinning OmegAMP's performance. It provides the fine-grained control over physicochemical properties that is necessary to move beyond unconditional generation and tackle the nuanced demands of creating peptides with complex target profiles, as we demonstrate next in Sec. 4.3.

### 4.3 TARGETED AMP GENERATION

Building upon OmegAMP's capabilities to adhere to pre-specified physicochemical properties reported in Tab. 4 and the conditioning scheme provided in Sec. 3.1, we leverage Subset Conditioning to generate sequences with improved efficacy against target bacteria, and Property Conditioning to reliably generate AMPs with target physicochemical patterns.

**Subset conditioning enables bacteria-specific AMP generation** To evaluate OmegAMP in the Subset Conditioning (SC) mode, we sample 10k peptides for multiple reference sets, including EV AMPs and sets of sequences known to be active against specific bacterial species. For the resulting bacteria-specific sequences and sequences from OmegAMP's unconditional mode, we compute the fraction of predicted positives according to our species-specific classifiers.

As shown in Fig. 2 a, subset conditioning drastically increases the fraction of predicted positives when compared to unconditional sampling. Notably, all subset-conditioned sequences achieve the highest AMP probabilities for their respective target classifiers (indicated by ticks). For instance, conditioning on sequences active against *A. Baumannii* allows the generation of peptides with an increased likelihood of efficacy against this species. From these results, we conclude that a smaller set

of high-quality conditioning vectors outperforms a larger set of mediocre ones, consider the smaller set of *A. Baumannii* (750 sequences) and the larger set of general EV AMPs (4209 sequences). For this species, the high-quality set achieves a more than two-fold improvement. Our results demonstrate that OmegAMP's generative model with subset conditioning reliably aligns generated peptides with desired activity profiles.

**Property conditioning closely follows physicochemical targets** To assess OmegAMP's Property Conditioning (PC), we sample 2k sequences, conditioned on specific target values for length (range 10-30) and three charge/hydrophobicity pairs. These pairs correspond to the (25th, 25th), (75th, 50th), and (25th, 75th) percentiles of their respective properties in our training set. For each target pair, we display the probability density of obtained charge and hydrophobicity values. We indicate each target pair by a colored cross.

Fig. 2 b shows that our model generates sequences that approximate the pre-specified constraints, such as a target pair of *charge=4* and *hydrophobicity=0.16* (colored in orange). Additionally, we observe a noticeable separation between the 3 regions, highlighting our model's ability to explore different antimicrobial clusters. These novel capabilities enable a new paradigm in AMP discovery, allowing practitioners to selectively explore specific physicochemical profiles.

**Property conditioning adheres to expert-defined intervals** Tab. 5 evaluates the capacity of various generative models to produce peptides within expert-defined physicochemical ranges associated with increased antimicrobial potential and synthesizability (charge: 2–10, length: 5–30, hydrophobicity: $-0.5$ to 0.8). We follow the experimental setting from Sec. 4.2 and compare OmegAMP variants against established baselines and a reference datasets (EV AMP). OmegAMP-PC exhibits state-of-the-art performance, satisfying individual constraints at rates exceeding 94% and achieving a combined success rate of 89.18%. Notably, this surpasses all baseline

Table 5: Comparison of generative models showing fraction of peptides meeting charge (C), length (L), hydrophobicity (H) and their simultaneous combination (C & L & H) criteria.

| Gen. Model | C | L | H | C & L & H |
|---|---|---|---|---|
| EV AMPs (*Data*) | 88.2 | 88.9 | 82.4 | 65.8 |
| AMP-GAN | 72.9 | 80.0 | 93.4 | 55.1 |
| Diff-AMP | 66.9 | **100.0** | 97.7 | 66.9 |
| HydrAMP | 58.3 | 99.9 | 89.4 | 50.5 |
| AMP-Diffusion | 62.3 | 62.2 | 94.4 | 39.1 |
| OmegAMP | 67.9 | 83.9 | 87.0 | 49.4 |
| OmegAMP-SC | 87.3 | 87.7 | 82.5 | 64.3 |
| OmegAMP-PC | **95.0** | 96.0 | **97.8** | **89.2** |

models suggesting superior capability in generating peptides with balanced and realistic properties. These findings underscore OmegAMP's ability to precisely adhere to complex design criteria, allowing practitioners to generate candidates that meet strict experimental requirements without relying on computationally expensive post-generation filtering.

## 4.4 WET-LAB VALIDATION

To validate OmegAMP's capabilities in a real-world setting, we performed an experimental evaluation of 25 peptides generated by our framework. These candidates were first designed using OmegAMP's conditional generative model and then prioritized for synthesis using our activity classifiers to maximize the likelihood of success. We synthesized these peptides and determined their Minimum Inhibitory Concentration (MIC) against a panel of clinically relevant pathogens. To see further details, we encourage the reader to take a look at App. K. To contextualize OmegAMP's performance, we compare our findings to previously reported experimental success rates of other approaches, including AMP-Diffusion, HydrAMP, CLaSS, and Joker (Torres et al., 2025b; Szymczak et al., 2023; Das et al., 2021; Porto et al., 2018). Success is measured as the cumulative fraction of peptides with activity at or below a given MIC threshold for at least one bacterial strain.

**OmegAMP delivers state-of-the-art success rate and potency** The experimental results, summarized in Fig. 3, show that OmegAMP outperforms all baseline methods. At the standard activity threshold of 32 $\mu$g/ml, OmegAMP achieved a near-perfect 96% success rate, substantially higher than AMP-Diffusion ($\sim$73%), HydrAMP ($\sim$58%), and other methods. More importantly, the results highlight the superior potency of OmegAMP-designed peptides. The success rate curve for OmegAMP rises sharply at very low MIC values, achieving an 80% success rate at just 4 $\mu$g/ml and over 90% at 8 $\mu$g/ml. In

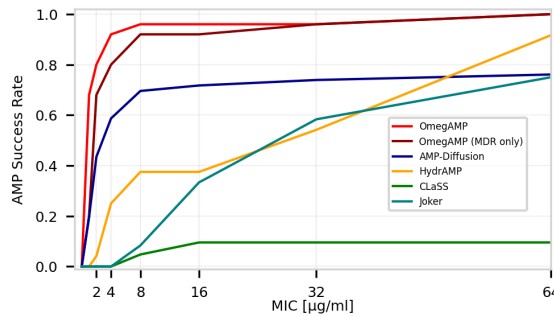

Figure 3: AMP success rate across various MIC thresholds for OmegAMP and baseline methods.

contrast, no baseline method surpassed a 40% success rate at these highly potent concentrations. This demonstrates that our framework not only generates active peptides but also candidates with strong, clinically relevant efficacy.

**High efficacy against multi-drug resistant (MDR) pathogens** A critical test for any AMP discovery platform is its ability to generate peptides effective against resistant pathogens. The performance of OmegAMP peptides tested exclusively against our panel of MDR strains was outstanding, achieving a success rate of 92% at 8 $\mu$g/ml. This result, nearly mirroring the overall success rate, confirms that OmegAMP does not overfit to non-resistant strains but effectively designs peptides capable of combating the most challenging pathogens.

## 5   CONCLUSION

In this work, we present OmegAMP, a principled framework for reliable conditional generation of AMPs. OmegAMP offers unprecedented control, enabling both species-specific peptide design and the generation of broad-spectrum antimicrobials. By incorporating diverse and effective conditioning mechanisms, it pushes the boundaries of controllable AMP generation, bringing computational design closer to real-world applications. Additionally, OmegAMP advances discriminator-guided filtering, leveraging a classifier that offers a substantial false positive rate reduction when compared to existing methods across multiple types of non-AMP sequences. The success of our computational framework was confirmed through wet-lab validation, where 24 out of 25 designed peptides (96%) demonstrated antimicrobial activity. These peptides proved to be highly potent, even against multi-drug resistant pathogens, bridging the gap between *in silico* design and tangible therapeutic candidates. These findings highlight OmegAMP's potential to accelerate the discovery of novel antimicrobial agents to combat the urgent threat of antimicrobial resistance.

## IMPACT STATEMENT

Our work on reliable conditional generation of AMPs has the potential to advance antimicrobial discovery, especially in low-data regimes. However, the ability to generate novel bioactive sequences could be misused to design harmful peptides. We do not intend for our research to be used in such a manner and encourage responsible applications aligned with public health and safety.

## LLM USAGE

This paper was written with the assistance of (Gemini-2.5, 2025), which was used to enhance language clarity and flow. All content has been reviewed and edited to ensure originality and accuracy.

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

## A    PRELIMINARIES

### A.1    DENOISING DIFFUSION PROBABILISTIC MODELS (DDPM)

Diffusion models (Sohl-Dickstein et al., 2015; Ho et al., 2020; Song et al., 2020) are powerful tools for approximating unknown data distributions, $p(\mathbf{x})$, by creating a mapping between a simple prior distribution, often chosen to be Gaussian, and the target data distribution. This process consists of two main steps: forward process and reverse process.

In the forward diffusion step, a data sample $\mathbf{x} \sim p(\mathbf{x})$, where $\mathbf{x} \in \mathbb{R}^d$, is transformed into a series of latent variables $\{\mathbf{z}_1, \ldots, \mathbf{z}_t, \ldots, \mathbf{z}_T\}$, where $t$ refers to time $t \in \{1, \ldots, T\}$. These variables progressively move from the data distribution towards the prior distribution over a sequence of timesteps. This transformation is modeled as a Markov chain, where noise is incrementally added at each step. For the Gaussian setting, the noise level is controlled by a variance schedule, $\beta_t$. Its cumulative product, $\alpha_t = \prod_{i=1}^{t} \beta_i$, determines the extent of perturbation applied at each timestep. The perturbed data $\mathbf{z}_t$ becomes:

$$\mathbf{z}_t = \sqrt{\alpha_t}\mathbf{x} + \sqrt{1 - \alpha_t}\epsilon,$$

where $\epsilon$ represents Gaussian noise, $\epsilon \sim \mathcal{N}(0, \mathbf{I})$, with $\mathbf{I}$ being the identity matrix.

In the reverse process, the noising of the forward process is inverted. Starting with pure Gaussian noise, $\mathbf{z}_T \sim \mathcal{N}(\mathbf{0}, \mathbf{I})$, the model learns to iteratively denoise the latent variables to reconstruct samples resembling the original data distribution.

To optimize this process, a loss function is used to minimize the reconstruction error:

$$L = \mathbb{E}_{\mathbf{x}, t, \mathbf{z}_t}\left[\left\|\hat{\mathbf{x}}_\theta(\mathbf{z}_t, t) - \mathbf{x}\right\|_2^2\right],$$

where $\hat{\mathbf{x}}_\theta$ represents the model's predicted reconstruction of the original data. Samples can be efficiently generated during the reverse process using various methods, such as DDPM (Ho et al., 2020) and DDIM (Song et al., 2020).

### A.1.1 CONDITIONING IN DDPM

In conditional models, additional information or context $c \in \mathbb{R}^{d'}$ can guide the generation process to produce samples with desired properties. For diffusion models, the conditioning information $c$ is incorporated into the reverse process by modifying the denoiser $\hat{\mathbf{x}}_\theta(\mathbf{z}_t, t)$ to also depend on $c$, resulting in $\hat{\mathbf{x}}_\theta(\mathbf{z}_t, t, c)$. In such models, $c$ can represent labels, attributes, or feature embeddings.

Notably, strong reliance on conditioning during sampling can lead to diversity loss or mode collapse due to peaked conditional distributions. To address this, Sadat et al. (2023) propose the *Condition-Annealed Diffusion Sampler* (CADS), which introduces controlled noise into the conditioning vector during sampling to balance diversity and specificity. The noised condition is defined as:

$$\hat{c}_t = \sqrt{\gamma(t)}c + s\sqrt{1 - \gamma(t)}\epsilon,$$

where $s$ controls the added noise, $\gamma(t)$ is the annealing schedule, and $\epsilon \sim \mathcal{N}(0, \mathbf{I})$. The annealing schedule $\gamma(t)$ transitions from 0 at $t \approx T$ (pure noise) to 1 at $t \approx 0$ (final denoising). During inference, the score function $\nabla_{\mathbf{z}_t} \log p_\theta(\mathbf{z}_t|\hat{c})$, which can be directly estimated using the denoising model $\hat{\mathbf{x}}_\theta(\mathbf{z}_t, t, \hat{c})$, smoothly shifts from the unconditional score $\nabla_{\mathbf{z}_t} \log p_\theta(\mathbf{z}_t)$ at high noise levels to the conditional term as the noise decreases. This approach allows CADS to maintain the diversity of unconditional models while effectively guiding the generation toward the desired conditioning properties.

## A.2 XGBOOST

While deep learning models have achieved significant success across diverse domains, Gradient Boosted Decision Trees (GBDTs), particularly implementations like XGBoost (Chen & Guestrin, 2016), remain highly competitive for supervised learning on tabular data, often demonstrating superior performance in various benchmarks compared to deep learning alternatives (Grinsztajn et al., 2022). We consider the application of XGBoost to binary classification. Given a dataset $\mathcal{D} = \{(x_i, y_i)\}_{i=1}^N$ where $x_i \in \mathbb{R}^M$ represents the feature vector for the $i$-th instance and $y_i \in \{0, 1\}$ is its corresponding binary label, the objective is to learn a predictive function. XGBoost constructs an ensemble model comprising $K$ additive regression trees (CARTs (Breiman, 2017)) belonging to a function space $\mathcal{F}$. The predicted probability for an instance $x_i$ is given by:

$$\hat{y}_i = \sigma\left(\sum_{k=1}^K f_k(x_i)\right), \quad f_k \in \mathcal{F}, \tag{4}$$

where each $f_k$ maps the input features $x_i$ to a continuous score associated with a leaf node in the $k$-th tree, and $\sigma(\cdot)$ is the sigmoid function. The functions $\{f_k\}_{k=1}^K$ are learned iteratively by minimizing the following regularized objective function:

$$\mathcal{L} = \sum_{i=1}^N l(y_i, \hat{y}_i) + \sum_{k=1}^K \Omega(f_k), \tag{5}$$

where $l(y_i, \hat{y}_i)$ is a differentiable loss function suitable for binary classification (e.g., logistic loss), and $\Omega(f_k)$ is a regularization term penalizing the complexity of the $k$-th tree (typically based on the number of leaves and the magnitude of leaf scores).

## B GENERATIVE MODEL DETAILS

### B.1 HYPERPARAMETER SELECTION & TRAINING REPRODUCIBILITY

To train our generative model we leveraged a NVIDIA gpu-GTX1080 with 8GB RAM. The model was trained for 72 hours. The hyperparameter details are presented in Tab. 6.

Table 6: Model Hyperparameters and Architecture Details

| Parameter | Value | Type |
|---|---|---|
| Beta Schedule | cosine | Diffusion Model |
| CADS-$\tau_1$ | 0.5 | Diffusion Model |
| CADS-$\tau_2$ | 0.9 | Diffusion Model |
| CADS-noise-scale | 0.1 | Diffusion Model |
| Timesteps | 1000 | Diffusion Model |
| Optimizer | radam | Diffusion Model |
| Batch Size | 128 | Diffusion Model |
| Learning Rate | 0.001 | Diffusion Model |
| LR Scheduler | Exponential Decay | Diffusion Model |
| Epochs | 1500 | Diffusion Model |
| Hidden Channels | 32 | Denoising Architecture |
| Downsampling Layers | 3 | Denoising Architecture |
| Downsampling Factor | 2 | Denoising Architecture |
| Upsampling Mode | nearest | Denoising Architecture |
| UNet Layer | Resnet Block + Linear Attention | Denoising Architecture |
| Total Parameters | 35,173,368 | Denoising Architecture |

## B.2 EMBEDDING SCHEME

### B.2.1 ENCODING

We define the encoding algorithm that transforms a peptide sequence into a fixed-size, residue-level embedding representation. Each amino acid in the input sequence is encoded as a $K$-dimensional vector using a predefined set of amino acid property scales $\{f_1, f_2, \ldots, f_K\}$. If the sequence length is shorter than the maximum length $M$, padding tokens are used to fill the remaining positions. For each position $i$ in the sequence (or padding), we compute the corresponding embedding vector $\boldsymbol{e}_i = [f_1(a), f_2(a), \ldots, f_K(a)]$, where $a$ is the amino acid or padding token at position $i$. The resulting matrix $\boldsymbol{E} \in \mathbb{R}^{K \times M}$ contains the full embedded representation of the input sequence. The full encoding process is formalized in Algorithm 1.

---

**Algorithm 1** Peptide Sequence Encoding

---

**Require:** Peptide sequence $\boldsymbol{s}$, maximum length $M$, amino acid scales $\{f_1, \ldots, f_K\}$
1: Initialize $\boldsymbol{E}$ as an $K \times M$ zero matrix
2: **for** $i = 1, \ldots, M$ **do**
3:    **if** $i \leq \text{length}(\boldsymbol{s})$ **then**
4:       $a \leftarrow \boldsymbol{s}[i]$
5:    **else**
6:       $a \leftarrow \text{PAD}$
7:    **end if**
8:    $\boldsymbol{e} \leftarrow [f_1(a), \ldots, f_K(a)]$
9:    $\boldsymbol{E}[:, i] \leftarrow \boldsymbol{e}$
10: **end for**
11: **return** $\boldsymbol{E}$

---

### B.2.2 DECODING

Following the proposed embedding scheme, we establish the decoding algorithm used to convert the embeddings back to the corresponding peptide sequences. As established before, for every amino-acid and padding token, we can compute a corresponding residue-level encoding $\boldsymbol{e}_i \leftarrow [f_1(a), f_2(a), \ldots, f_K(a)]$, therefore we can map embeddings by iteratively finding the closest encoding within the 21 possible encodings (20 amino-acids + padding token). Additionally, for simplicity, we terminate decoding after encountering a padding token. We formalize our decoding process in Algorithm 2.

---

**Algorithm 2** Embedding Decoding

---

**Require:** Embedding $E$, maximum length $M$, amino acid scales $\{f_1, f_2, \ldots, f_K\}$

1: $s \leftarrow \emptyset$
2: **for** $i = 1, \ldots, M$ **do**
3:    $a \leftarrow \arg\min_{a' \in \mathcal{A} \cup \{\text{PAD}\}} \|E[:, i] - [f_1(a'), \ldots, f_K(a')]\|_2$
4:    **if** $a \neq \text{PAD}$ **then**
5:       $s \leftarrow s \cup \{a\}$
6:    **else**
7:       **return** $s$
8:    **end if**
9: **end for**
10: **return** $s$

---

## C  DATASETS

In this section we describe the datasets used for the training of generative model and the classifiers in OmegAMP.

**Generative Dataset**   The generative dataset comprises a diverse collection of AMP and general peptide sequences from well-established databases of length at most 100:

- **AMP sequences**: 36,262 sequences from AMPScanner (Veltri et al., 2018), dbAMP (Jhong et al., 2022), and DRAMP (Shi et al., 2022).

- **General peptide sequences**: 774,405 sequences from Peptipedia (Cabas-Mora et al., 2024), consisting of functional peptides extracted from Uniprot (Consortium, 2022) that were classified as Antibacterial, Anti Gram + or Anti Gram - by Peptipedia's prediction algorithms.

This dataset provides a broad representation of peptide sequences for training the generative model. We deliberately select peptides that are similar to AMPs to incentivize the generative model to learn meaningful activity patterns, while retaining scientific rigor by representing them in distinct groupings. Although some labeling noise may arise from discrepancies in source databases, the dataset's scale is essential for learning robust sequence representations.

**Classifier Datasets**   Classifiers in OmegAMP are trained on two crucial data sources: Experimentally Verified (EV) datasets and a non-EV component, ensuring a reliable basis for training and evaluation.

To account for potency and specificity of AMPs, we construct a set of high quality datasets which consist of experimentally validated peptides with known activity values against target microbes. To this purpose, peptide sequences together with their Minimal Inhibitory Concentration (MIC) measurements were downloaded from DBAASP database (Pirtskhalava et al., 2021). We exclude sequences which contain non-standard amino acids, and sequences with non-standard C- and N-terminus. We further standardize the experimental conditions with respect to the medium and colony forming unit (CFU).

For the general AMP/non-AMP classification we consider as positives peptides with MIC $\leq 32\mu g/\text{mL}$ against at least one bacterial strain, and as negatives sequences with MIC $\geq 128\mu g/\text{mL}$ for all strains.

For the strain- and species- specific classification, we select peptides, which were experimentally proven to show activity against the microbes of interest. A peptide is considered as active (positive) against a specific strain or species if its MIC $\leq 32\,\mu g/\text{mL}$ and inactive (negative) if its MIC $\geq 128\,\mu g/\text{mL}$.

Additionally, the non-EV consists exclusively of non-AMPs and is composed by a dataset of non-AMPs from AMPlify (Li et al., 2022), coupled with 100k synthetic sequences per each source (random, shuffled, mutated), see Sec. 3.2 for further details.

The resulting dataset composition is summarized in Tab. 7.

Table 7: Dataset statistics for classifier training, grouped by data source. EV stands for Experimentally Validated

| EV | Dataset Group | Positives | Negatives |
|---|---|---|---|
| Yes | General | 4209 | 920 |
| | Species - *A. baumannii* | 750 | 243 |
| | Species - *E. coli* | 2939 | 1086 |
| | Species - *K. pneumoniae* | 685 | 421 |
| | Species - *P. aeruginosa* | 1632 | 935 |
| | Species - *S. aureus* | 2385 | 1230 |
| | Strain - *A. baumannii* (ATCC 19606) | 313 | 105 |
| | Strain - *E. coli* (ATCC 25922) | 1671 | 541 |
| | Strain - *K. pneumoniae* (ATCC 700603) | 278 | 121 |
| | Strain - *P. aeruginosa* (ATCC 27853) | 825 | 423 |
| | Strain - *S. aureus* (ATCC 25923) | 988 | 423 |
| | Strain - *S. aureus* (ATCC 33591) | 60 | 58 |
| | Strain - *S. aureus* (ATCC 43300) | 278 | 106 |
| No | AMPlify's non-AMPs | – | 127,983 |
| | Synthetic Random | – | 100,000 |
| | Synthetic Shuffled | – | 100,000 |
| | Synthetic Mutated | – | 100,000 |

It is important to note that the non-EV component remains the same across all classification tasks, while the EV dataset varies depending on the task. For instance, for the classifier that determines sequences active against *A. baumannii* we have only 750 EV positives and 243 EV negatives.

## D  CLASSIFIER DETAILS

### D.1  CLASSIFIER FEATURES

Our XGBoost classifiers are trained on a comprehensive set of 276 features engineered to capture a wide range of physicochemical and sequence-based properties critical for antimicrobial activity. These features are derived directly from the peptide sequences and can be grouped into three main categories.

**Global Peptide Descriptors**  We compute a total of 156 global descriptors that summarize the overall physicochemical characteristics of a peptide sequence. This set includes fundamental properties such as molecular weight, peptide length, isoelectric point (pI), aromaticity, and instability index. Additionally, we incorporate multiple metrics for charge and hydrophobicity, which are known to be key drivers of antimicrobial function. These features provide a sequence-level summary of the peptide. For the calculation of these descriptors, we rely on established and widely-used bioinformatics libraries, including Biopython (Chapman & Chang, 2000), modlAMP (Müller et al., 2017), Peptides (Larralde, 2021), and Peptidy (Özçelik et al., 2025).

**Amino Acid Composition**  To provide an overview of a peptide's makeup, we include a set of 20 features representing the amino acid composition. This is calculated as the relative frequency (fraction) of each of the 20 standard amino acids within a given sequence. This feature set offers a permutation-invariant view of the building blocks of the peptide, which is essential for distinguishing between peptides with different residue preferences.

**Exponential Moving Average of Amino Acid Scales**  To capture sequential information and local physicochemical context without the overhead of complex sequence models, we introduce a set of features based on the Eisenberg Amino-Acid Scale (Eisenberg et al., 1984). To this end, we use the Exponential Moving Average (EMA) of the residues in the forward direction, i.e., from beginning to end, to provide features that can be easily used in a decision tree setting. We provide features for the first 100 positions.

## D.2 Hyperparameter Selection & Training Reproducibility

To train our classifiers, we utilized a regular CPU, and the training took approximately 30 minutes. The hyperparameter details are presented in Tab. 8.

Table 8: Classifier Hyperparameters

| Parameter / Setting | Value / Configuration |
|---|---|
| Classifier Type | XGBoost |
| Maximum Estimators | 5000 |
| Maximum Tree Depth | 6 |
| Early Stopping | Enabled |
| Patience | 50 rounds |
| Validation Set Size | 3% of training data |

## D.3 Loss Function

As introduced in the main text, our weighted binary cross-entropy loss addresses both data quality and class imbalance. For a training set containing $N^0$ non-AMP and $N^1$ AMP sequences, each sample's contribution is adjusted by a weight function $\omega(\boldsymbol{s})$ that factors in the data source:

$$\omega(\boldsymbol{s}) = \omega_1 \mathbb{I}_{\{\boldsymbol{s} \text{ is EV}\}} + \omega_0 \mathbb{I}_{\{\boldsymbol{s} \text{ is not EV}\}} \quad . \tag{6}$$

Here, the terms $\omega_1 := \frac{N^0 + N^1}{2N^1}$ and $\omega_0 := \frac{N^0 + N^1}{2N^0}$ are standard class-balancing weights that ensure equal total contribution from the positive and negative classes, respectively.

This formulation effectively prioritizes the high-confidence EV data. Since our positive set consists entirely of EV sequences and the negative set is dominated by non-EV synthetic data (see App. C), the ratio $\frac{N^1}{N^0}$ is small. Consequently, the weight $\omega_1$ assigned to EV samples is significantly larger than the weight $\omega_0$ assigned to non-EV samples, focusing the model's learning on experimentally verified examples.

# E Amino-Acid Scales

Amino acid scales represent the physicochemical properties of amino acids often derived from biochemical experiments (Wilce et al., 1995). It is important to note that no consensus exists on the optimality of any single scale (Simm et al., 2016). This lack of consensus is expected because these scales are derived from distinct biochemical experiments and numerical methods. Consequently, the usefulness of each scale is closely tied to its specific application.

An important observation is that not all scales provide an injective mapping. For example, the Kyte-Doolittle scale assigns the same value $(-3.5)$ to Asparagine, Aspartic Acid, Glutamine, and Glutamic Acid. This leads to unsuitable mappings when invertibility is required, as injectivity is a necessary property for the existence of an inverse function.

For completeness, we provide a table with the scales used in the embedding scheme (see Tab. 9).

Table 9: Amino-acid scales utilized in the embedding scheme. WW* defines a slightly altered version of the Wimley-White scale, and TM reads Transmembrane Propensity scale.

| AA | WW* | pI | Levitt | TM | AASI |
|----|------|-------|--------|--------|------|
| A | -0.03 | 6.01 | 1.290 | 11.200 | 1.89 |
| R | -0.74 | 10.76 | 0.960 | 0.500 | 1.91 |
| N | -0.28 | 5.41 | 0.900 | 2.900 | 2.33 |
| D | -1.23 | 2.85 | 1.040 | 2.900 | 3.13 |
| C | 0.71 | 5.05 | 1.110 | 4.100 | 1.73 |
| Q | -0.51 | 5.65 | 1.270 | 1.600 | 3.05 |
| E | -2.02 | 3.15 | 1.440 | 1.800 | 3.14 |
| G | 0.37 | 6.06 | 0.560 | 11.800 | 2.67 |
| H | -0.89 | 6.00 | 1.220 | 2.000 | 3.00 |
| I | 0.81 | 6.05 | 0.970 | 8.600 | 1.97 |
| L | 1.06 | 6.01 | 1.300 | 11.700 | 1.74 |
| K | -0.99 | 9.60 | 1.230 | 0.500 | 2.28 |
| M | 0.61 | 5.74 | 1.470 | 1.900 | 2.50 |
| F | 1.63 | 5.49 | 1.070 | 5.100 | 1.53 |
| P | -0.38 | 6.30 | 0.520 | 2.700 | 0.22 |
| S | 0.17 | 5.68 | 0.820 | 8.000 | 2.14 |
| T | 0.07 | 5.60 | 0.820 | 4.900 | 2.18 |
| W | 2.35 | 5.89 | 0.990 | 2.200 | 2.00 |
| Y | 1.44 | 5.64 | 0.720 | 2.600 | 2.01 |
| V | 0.27 | 6.00 | 0.910 | 12.900 | 2.37 |

## F    SPECIES/STRAIN ACTIVITY

For clarity and completeness, we provide a formal definition for species/strain activity. In particular, we claim that a peptide is active against a bacterial strain if it satisfies the following definition:

**Definition 1.** A peptide $s \in \mathcal{A}^L$ is considered active against a bacterial strain $b \in \mathcal{B}$ if and only if $\mathrm{MIC}(b, s) \leq 32\mu\mathrm{g/mL}$.

Additionally, we state that a peptide is active against a specific bacterial species if it is active against at least one strain of that species.

**Definition 2.** A peptide $s \in \mathcal{A}^L$ is considered active against a bacterial species if and only if it demonstrates activity against at least one strain of that species $\{b_1, b_2, \ldots\} \subseteq \mathcal{B}$. Formally, $\exists b \in \{b_1, b_2, \ldots\} : \mathrm{MIC}(b, s) \leq 32\mu\mathrm{g/mL}$.

In these definitions, we use the Minimum Inhibitory Concentration (MIC), which is defined as the lowest peptide concentration needed to inhibit visible bacterial growth under standard experimental conditions. Moreover, we utilize the $32\mu\mathrm{g/mL}$ threshold because of its prominence in various experimental work (Torres et al., 2025a; Szymczak et al., 2023). Finally, we note that when inferring species-specific activity, there exists an implicit assumption on the considered set of bacterial strains, since, due to practical limitations, we often don't have data for all known strains.

## G    INACTIVITY OF SYNTHETIC DATA

**Theoretical Motivation**    Let $\mathcal{S}_L^1 = \{s \in \mathcal{A}^L \mid y = 1\}$ represent the set of AMP sequences of length $L$. The probability of a random sequence $s \in \mathcal{A}^L$ being an AMP is:

$$\mathbb{P}(y = 1 \mid s) = \frac{|\mathcal{S}_L^1|}{|\mathcal{A}|^L}.$$

As shuffling an AMP typically disrupts its activity (Porto et al., 2022), we can estimate:

$$\mathbb{P}(y = 1 \mid s) \leq \frac{1}{|\{\pi(s) \mid \pi \in \mathcal{P}_L\}|} \approx \frac{1}{L!},$$

where $\mathcal{P}_L$ is the symmetric group of permutations of $L$ elements, and $\pi(\boldsymbol{s})$ represents the application of a permutation $\pi$ to the sequence $\boldsymbol{s}$. Note that this upper bound is rather generous as it implicitly assumes that every sequence can be shuffled into an AMP, which is highly unlikely. For $N$ sampled sequences $\boldsymbol{s}$ of length $L$, where each sequence $\boldsymbol{s} = (a_1, a_2, \ldots, a_L)$ consists of i.i.d. amino acids $a_i$ drawn from the uniform distribution over the amino acid space $\mathcal{A}$, $a_i \sim U(\mathcal{A})$, the expected number of AMPs is bounded by:

$$\mathbb{E}\Big[\sum_{i=1}^{N} X_i\Big] = N \cdot \mathbb{P}(y = 1 \mid \boldsymbol{s}) \leq \frac{N}{L!},$$

where $X_i \sim \mathrm{Bernoulli}(\mathbb{P}(y = 1 \mid \boldsymbol{s}))$. These observations imply that for $N \approx 10^6$ and large $L$, i.e. $L > 10$, the expected number of AMPs is small, strengthening the claim that synthetic sequences are expected to be inactive.

**Empirical Motivation**     Prior research has shown that antimicrobial sequences constitute a small fraction (less than 5%, even with lenient definitions) of both random (Tucker et al., 2018) and shuffled sequences (Porto et al., 2022; Loose et al., 2006). Furthermore, to see that there exists a distribution shift between active sequences and synthetic sequences, we evaluate the OmegAMP classifier and other external classifiers on the external dataset from Tucker et al. (2018) in App. J.4. Despite the difference of AMP criteria between datasets, we see classifiers consistently displaying a statistically significant ability to distinguish active from synthetic sequences, thus strengthening the claim that these sets are separable by a decision boundary, which, given the binary nature of the problem, implies that the synthetic sequences are likely inactive.

We also investigate the physicochemical distributions and other key metrics across sequences from the EV AMP dataset, the EV non-AMP dataset, and synthetic sets including random, shuffled, and mutated sequences, as illustrated in Fig. 4. We find noticeable distribution shifts in Fitness Score and Pseudo Perplexity when comparing natural EV AMPs to synthetic sequences. Specifically, AMPs consistently demonstrate significantly higher fitness scores than all categories of synthetic sequences (random, shuffled, and mutated). This distinction is particularly important as higher fitness scores are known indicators of peptide functionality and antimicrobial activity. The synthetic sequences often fall outside these optimal ranges, suggesting they occupy regions of sequence space less likely to display antimicrobial properties. Furthermore, we observe distinct patterns in physicochemical properties such as charge and hydrophobicity, which further differentiate AMPs from random and mutated sequences. Higher Pseudo Perplexity of synthetic sequences indicates that they are less biologically plausible. These differences strongly suggest that generating sequences randomly, through shuffling or by mutating original AMPs is unlikely to produce AMPs.

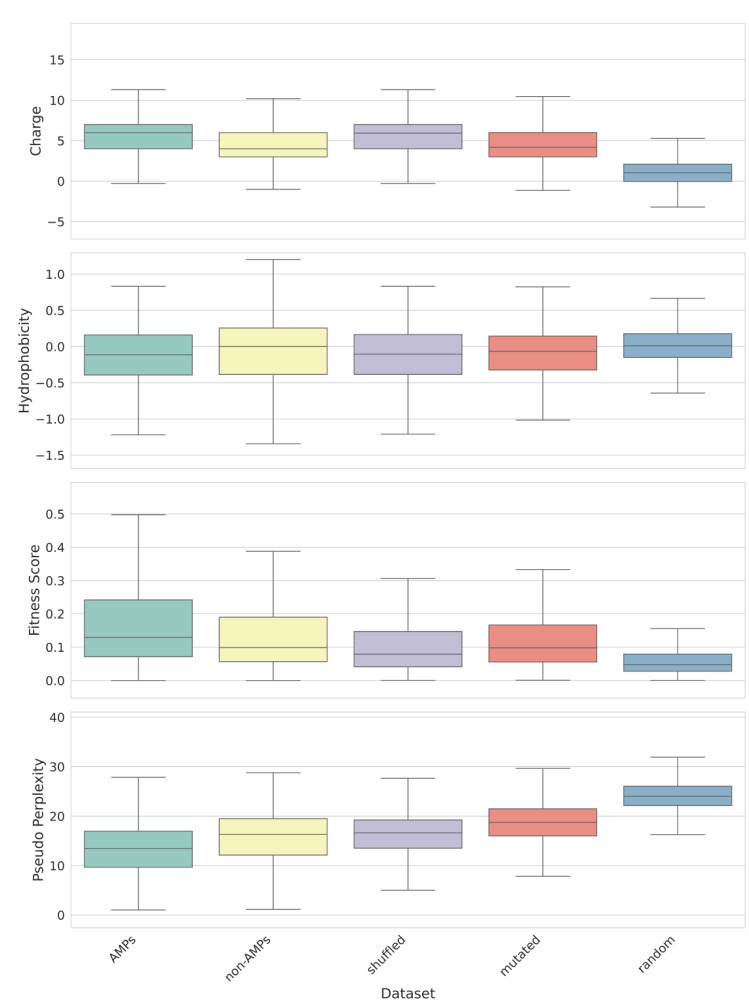

Figure 4: Empirical distributions of physicochemical (charge, hydrophobicity) and model-derived (fitness score, pseudo perplexity) characteristics for natural EV AMPs, EV non-AMPs, and synthetic sequences. Natural AMPs display higher fitness scores and lower pseudo perplexity when compared to other synthetic sequences.

## H    METRICS

In the main text, we utilize various metrics to assess the quality of generated peptides. Here, we provide a detailed explanation of how these metrics are computed for a set of sequences $\mathcal{S} := \{s_i\}_{i=1}^N$.

**Diversity**   We compute this metric by calculating the average normalized alignment between two peptides. Alignment can be defined as the largest possible subset of ordered characters that both sequences share, as originally proposed in Needleman & Wunsch (1970).

$$\text{Diversity}(\mathcal{S}) = \frac{100}{\binom{N}{2}} \times \sum_{s_i \in \mathcal{S}} \sum_{s_j \in \mathcal{S} \setminus \{s_i\}} \frac{\text{Alignment}(s_i, s_j)}{\min(\text{length}(s_i), \text{length}(s_j))}$$

**Uniqueness**   To obtain the Uniqueness of a set of sequences, we compute the percentage of distinct sequences within the set.

$$\text{Uniqueness}(\mathcal{S}) = \frac{|\{s_i | s_i \notin \{s_1, \dots, s_{i-1}\}\}|}{N} \times 100$$

Table 10: Hydrophobicity and helix propensity scales used in fitness calculation.

| AA | h | hx |
|----|------|------|
| A | 0.25 | 0.00 |
| R | -1.80 | 0.21 |
| N | -0.64 | 0.65 |
| D | -0.72 | 0.69 |
| C | 0.04 | 0.68 |
| Q | -0.69 | 0.39 |
| E | -0.62 | 0.40 |
| G | 0.16 | 1.00 |
| H | -0.40 | 0.61 |
| I | 0.73 | 0.41 |
| L | 0.53 | 0.21 |
| K | -1.10 | 0.26 |
| M | 0.26 | 0.24 |
| F | 0.61 | 0.54 |
| P | -0.07 | 3.16 |
| S | -0.26 | 0.50 |
| T | -0.18 | 0.66 |
| W | 0.37 | 0.49 |
| Y | 0.02 | 0.53 |
| V | 0.54 | 0.61 |

**Novelty**   To obtain the novelty of a set of sequences, we compute the number of non-overlapping sequences between the aforementioned set of sequences and the EV AMPs, which we denote with $\mathcal{H}$.

$$\text{Novelty}(\mathcal{S}) = \frac{|\{s_i | s_i \notin \mathcal{H}|}{N} \times 100$$

**Fitness Score**   Let $\theta := \frac{100 \times \pi}{180}$, i.e. the equivalent of 100 degrees in radians, and both $h$ and $hx$ denote the amino-acid scales presented in Tab. 10, then we can define the Fitness-Score (Li et al., 2024).

$$\text{Fitness-Score}(\mathcal{S}) = \frac{1}{N} \times \sum_{(a_1,\ldots,a_L) \in \mathcal{S}} \frac{\sqrt{\left(\sum_{i=1}^{L} h(a_i)\cos(i\theta)\right)^2 + \left(\sum_{i=1}^{L} h(a_i)\sin(i\theta)\right)^2}}{\sum_{i=1}^{L} e^{hx(a_i)}}$$

**Pseudo-Perplexity**   Let $p_\phi$ denote a density estimator for 1-amino-acid masked language modelling, which in our case consists of ESM2 (Lin et al., 2022). Then, we can define the Pseudo-Perplexity metric as follows:

$$\text{Pseudo-Perplexity}(\mathcal{S}) = \frac{1}{N} \times \sum_{(a_1,\ldots,a_L) \in \mathcal{S}} \exp\left\{-\frac{1}{L} \sum_{i=1}^{L} \log p_\phi\big(a_i \mid a_1,\ldots,a_{i-1}, a_{i+1},\ldots,a_L\big)\right\}$$

# I   FURTHER EXPERIMENTS GENERATION

## I.1   GENERATIVE DATASET ABLATION

To quantify the impact of including the large-scale "General Peptide Sequences" from Peptipedia in our training data, we performed an ablation study where we trained the generative model exclusively on the curated "AMP sequences" (approx. 36k sequences), excluding the 774k general peptides, see App. C for dataset details. We evaluated the models on unconditional generation metrics and conditional controllability. These metrics are described in Sec. 4.2.

As shown in Tab. 11, removing the general peptide sequences results in a degradation of performance across key metrics. Specifically, the model trained on the full dataset (OmegAMP) achieves a higher OmegAMP classification rate (10.5% vs 8.1%) and a higher Fitness Score (0.13 vs 0.11), indicating better biological plausibility. Furthermore, the full model demonstrates superior controllability, with significantly lower Mean Absolute Errors (MAE) for Length (0.04 vs 0.23), Charge (0.16 vs 0.27), and Hydrophobicity (0.18 vs 0.27). These results confirm that exposing the model to a larger, chemically diverse set of peptide sequences is crucial for learning robust representations that facilitate both high-quality generation and precise physicochemical control.

Table 11: Ablation study assessing the impact of including the general peptide sequences to the training set. We compare the full OmegAMP model against a variant trained without the large-scale general peptide sequences. The evaluation utilizes the metrics presented in Sec. 3.1

| Gen. Model | Omeg. Class. | Fit. Score | Div. | Uniq. | MAE (in std units) (↓) | | |
| --- | --- | --- | --- | --- | --- | --- | --- |
| | | | | | Length | Charge | Hydroph. |
| OmegAMP w/o General Peptides | 8.1 | 0.11 | 0.60 | **96** | 0.23 | 0.27 | 0.27 |
| OmegAMP | **10.5** | **0.13** | **0.64** | 94 | **0.04** | **0.16** | **0.18** |

## I.2 Embedding Quality Comparison

To compare the quality of the OmegAMP biologically-informed embedding against established protein language model representations, we compared it with ESM-2 embeddings (Lin et al., 2022). We trained XGBoost regressors to predict five key physicochemical properties (Charge, Hydrophobicity, Instability Index, Boman Index, and Aliphatic Index) using different embedding schemes as input features. We utilized the "AMP Sequences" subset of the generative dataset, see App. C, and applied a 5-fold cross-validation scheme.

Tab. 12 reports the Mean Absolute Error (MAE) for each property. The OmegAMP embedding consistently achieves the lowest MAE across all tested properties compared to both PCA-reduced and Average-Pooled ESM-2 embeddings. For instance, the MAE for Charge prediction is 0.526 for OmegAMP versus 0.556 for ESM-2 (Avg. Pooling). This suggests that our compact, biologically-inspired embedding provides an explicit and chemically rich representation where physicochemical properties are more linearly separable and easier to decode than in the larger, generic latent spaces of protein language models.

Table 12: Performance of different peptide embeddings on predicting physicochemical properties. Lower MAE indicates a representation that better captures the underlying physicochemical attributes.

| Embedding Type | MAE (avg ± std) (↓) | | | | |
| --- | --- | --- | --- | --- | --- |
| | Charge | Hydrophobicity | Instability Index | Boman Index | Aliphatic Index |
| ESM-2 (PCA) | $0.683 \pm 0.007$ | $0.084 \pm 0.001$ | $19.55 \pm 0.17$ | $0.448 \pm 0.005$ | $12.06 \pm 0.09$ |
| ESM-2 (Avg. Pooling) | $0.556 \pm 0.004$ | $0.067 \pm 0.000$ | $18.30 \pm 0.16$ | $0.359 \pm 0.005$ | $10.20 \pm 0.08$ |
| OmegAMP | $\mathbf{0.526 \pm 0.010}$ | $\mathbf{0.063 \pm 0.001}$ | $\mathbf{15.70 \pm 0.21}$ | $\mathbf{0.328 \pm 0.004}$ | $\mathbf{8.12 \pm 0.15}$ |

## I.3 Amino-Acid Frequency Comparison

For further analysis of generated peptides we provide the amino-acid frequencies for all considered generative models, see Fig. 5. The amino acid frequency distribution of unconditional OmegAMP-generated sequences closely aligns with that of the AMP training data. This alignment suggests that our generative model effectively captures key sequence-level features characteristic of AMPs, and can produce biologically relevant candidates.

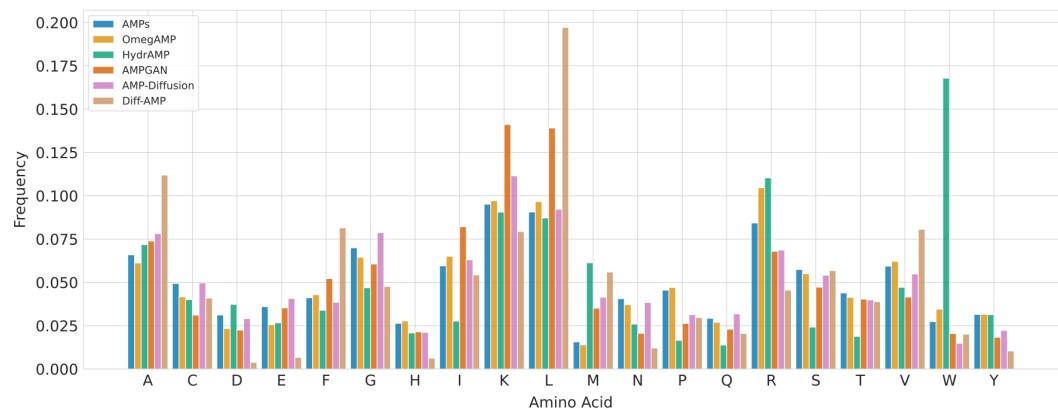

Figure 5: Amino acid frequency distribution comparison between OmegAMP-generated sequences and AMP training data. The close alignment shows that OmegAMP captures key AMP sequence features, ensuring biologically relevant generation.

## I.4 ON CONTROLLING MULTIPLE PROPERTIES SIMULTANEOUSLY

Following the experimental setup of the single-property controllability evaluation Sec. 4.2, we assess OmegAMP's ability to control multiple physicochemical properties simultaneously. We conditioned the model to generate peptides adhering to target values for all three properties—Charge, Length, and Hydrophobicity—at once. The results, presented in Tab. 13, compare the Mean Absolute Error (MAE) for both single- and multi-property conditioning. The MAEs remain low in the multi-property setting, showing only a slight increase from the single-property baseline. This demonstrates that OmegAMP can robustly handle the more challenging multi-property conditioning task without a significant drop in performance.

Table 13: Mean Absolute Errors (MAEs) in standardized units for different conditioning modes of OmegAMP. Lower values indicate better performance.

| Conditioning Mode | MAE (in std units) (↓) | | |
|---|---|---|---|
| | Length | Charge | Hydroph. |
| OmegAMP Single-Property | 0.04 | 0.16 | 0.18 |
| OmegAMP Multi-Property | 0.27 | 0.20 | 0.20 |

To further visualize OmegAMP's ability to precisely target specific physicochemical regions, we performed a grid sweep analysis. We conditioned the model on a Cartesian product grid of Target Charge values $\{0, 2, 4, 6, 8, 10\}$ and Target Hydrophobicity values $\{-0.5, -0.2, 0.0, 0.2, 0.4, 0.6, 0.8\}$, sampling 250 sequences per pair.

Fig. 6 displays the deviation (Mean Absolute Error) between the target and generated properties. The heatmaps demonstrate that OmegAMP maintains high controllability (low deviation, indicated by lighter colors) across the majority of the biologically feasible landscape. Higher deviations are observed in biochemically constrained regions where satisfying both properties simultaneously is physically difficult (e.g., extremely high charge combined with high hydrophobicity).

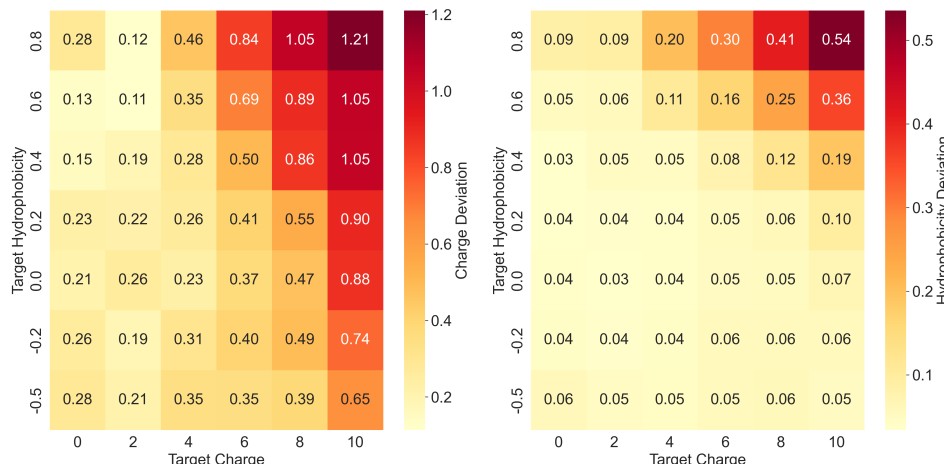

Figure 6: Heatmaps illustrating conditional controllability. The plots show the Mean Absolute Error between target and generated values for Charge (left) and Hydrophobicity (right) across a defined target grid.

## I.5 ANALYSIS OF CONDITIONING STRATEGIES: PC VS. SC

In the main text, we observed that Property Conditioning (PC) using expert-defined ranges often yields higher classifier scores than Subset Conditioning (SC). To investigate whether this is due to the ranges selected or the conditioning method itself, we evaluated PC using the exact same property ranges inherent to the SC exemplar sets (which are typically much wider and include "low-hit-rate" zones).

Tab. 14 shows that when PC is forced to sample from the broader, less targeted ranges used in SC (e.g., Length 2-98 vs expert 10-30), its performance drops significantly (OmegAMP Class. 3.2% vs 14.8%). This confirms that the superior performance of PC in our main results is driven by the explicit guidance provided by expert knowledge, which directs the model toward high-probability activity zones. Conversely, SC is valuable when specific distributional correlations of a target species are needed, even if it involves sampling from harder-to-model regions.

Table 14: Performance comparison investigating the impact of property ranges on conditioning strategies.

| Gen. Model | HydrAMP-MIC | OmegAMP Class. | Fitness Score | Diversity | Uniqueness | Novelty |
|---|---|---|---|---|---|---|
| OmegAMP | 33.8 | 10.5 | 0.13 | **0.64** | 94 | 98 |
| OmegAMP-PC (SC Property Ranges) | 40.6 | 3.2 | 0.12 | 0.57 | 93 | **100** |
| OmegAMP-PC (Expert Ranges) | **70.2** | 14.8 | **0.16** | 0.60 | **98** | 99 |
| OmegAMP-SC | 64.1 | **16.4** | 0.15 | 0.61 | 95 | 97 |

## J FURTHER EXPERIMENTS CLASSIFICATION

### J.1 SPECIES/STRAIN SPECIFIC CLASSIFICATION

We further evaluate species- and strain-specific classifiers as outlined in Sec. 3.2 to test whether the performance of our general classifier, see Tab. 1, generalizes to narrower biological contexts with limited training data.

To accomplish this, we train strain- and species- specific classifiers on their respective dataset, which we detail in App. C. Otherwise, we follow the experimental setup as described in Sec. 4.1. The results in Tab. 15 demonstrate that OmegAMP's strong performance generalizes to species- and strain-specific classification. Despite fewer training samples, these specialized models show TPR, FPR, and LR+ values comparable or superior to the general classifier (Tab. 1), with LR+ reaching up to 28488.5.

Table 15: Performance metrics for species and strain-specific OmegAMP classifiers. Columns are as in Tab. 1.

| Target | Performance Metrics | | | | | Robustness (Misclassification Rate %) | | | | | |
| | | | | | | Bio Non-AMPs | | Synthetic Non-AMPs | | | |
| | AUPRC (↑) | Prec@100 (↑) | TPR (↑) | FPR (↓) | LR+ (↑) | Sig | Met | AD | R | S | M |
|---|---|---|---|---|---|---|---|---|---|---|---|
| *A. baumannii* | 47.6 | 75.7 | 33.5 | 0.0 | 762.1 | 0.0 | 0.1 | 0.1 | 0.0 | 0.1 | 0.1 |
| *E. coli* | 61.4 | 95.2 | 44.9 | 0.2 | 289.8 | 0.0 | 0.1 | 0.3 | 0.0 | 0.3 | 0.6 |
| *K. pneumoniae* | 58.3 | 87.7 | 39.9 | 0.0 | 2017.7 | 0.0 | 0.0 | 0.0 | 0.0 | 0.1 | 0.2 |
| *P. aeruginosa* | 54.6 | 88.4 | 37.9 | 0.1 | 562.8 | 0.0 | 0.1 | 0.2 | 0.0 | 0.2 | 0.3 |
| *S. aureus* | 53.4 | 86.2 | 38.2 | 0.1 | 259.0 | 0.0 | 0.2 | 0.2 | 0.0 | 0.4 | 0.4 |
| *A. baumannii* ATCC19606 | 54.7 | 85.5 | 39.5 | 0.0 | 3594.2 | 0.0 | 0.0 | 0.0 | 0.0 | 0.0 | 0.1 |
| *E. coli* ATCC25922 | 56.9 | 89.8 | 39.6 | 0.1 | 460.0 | 0.0 | 0.1 | 0.2 | 0.0 | 0.2 | 0.3 |
| *K. pneumoniae* ATCC700603 | 64.6 | 92.5 | 47.0 | 0.0 | 8551.6 | 0.0 | 0.0 | 0.0 | 0.0 | 0.1 | 0.1 |
| *P. aeruginosa* ATCC27853 | 58.9 | 87.7 | 41.5 | 0.0 | 1608.3 | 0.0 | 0.0 | 0.1 | 0.0 | 0.1 | 0.1 |
| *S. aureus* ATCC25923 | 48.5 | 79.9 | 32.2 | 0.0 | 761.6 | 0.0 | 0.0 | 0.1 | 0.0 | 0.2 | 0.1 |
| *S. aureus* ATCC33591 | 23.6 | 70.0 | 15.7 | 0.0 | 28488.5 | 0.0 | 0.0 | 0.0 | 0.0 | 0.0 | 0.0 |
| *S. aureus* ATCC43300 | 47.2 | 80.3 | 31.7 | 0.0 | 2750.7 | 0.0 | 0.0 | 0.0 | 0.0 | 0.1 | 0.1 |

Furthermore, their prediction rates for synthetic negatives are consistently lower that those of the baselines. This improved discriminative ability highlights OmegAMP's practical utility for precise predictions across diverse biological contexts.

## J.2 SYNTHETIC DATA ABLATION

To validate the inclusion of each type of synthetic sequences (Random, Shuffled, and Mutated) to the training set presented in Sec. 3.2, we perform an ablation study that analyzes these contributions separately. Apart from the implied adjustments to the training, the experimental setup is identical to that of Sec. 4.1.

Tab. 16 demonstrates that including specific synthetic datasets into the training yields improvement not only across the respective synthetic probabilities, but also for types of inactive sequences unseen during training, namely Signal and Metabolic peptides, as well as the more challenging Added-Deleted Synthetics. In summary, when compared with its ablations, OmegAMP displays the highest AUPRC, Prec@100, and LR+ of 56.9, 90.4%, and 138.1, respectively. These improvements, along with a drastically reduced misclassification rate for unseen inactive types, suggest that synthetic sequence augmentation and a weighted loss function aid our classifier in learning genuine sequence-function relationships rather than spurious correlations —a conclusion supported by our interpretability analysis in App. J.5— thereby enhancing its ability to generalize to unseen antimicrobial peptides.

Table 16: Performance Metrics for OmegAMP ablations with varying integration of synthetic negatives and loss formulations. Columns are as in Tab. 1.

| Model | Performance Metrics | | | | | Robustness (Misclassification Rate %) | | | | | |
| | | | | | | Bio Non-AMPs | | Synthetic Non-AMPs | | | |
| | AUPRC (↑) | Prec@100 (↑) | TPR (↑) | FPR (↓) | LR+ (↑) | Sig | Met | AD | R | S | M |
|---|---|---|---|---|---|---|---|---|---|---|---|
| Omeg. − {R, S, M} | 19.0 | 36.2 | **95.9** | 5.7 | 3.5 | 4.6 | 6.6 | 85.6 | 5.7 | 95.1 | 77.4 |
| Omeg. − {S, M} | 27.1 | 52.6 | 93.2 | 21.9 | 4.3 | 4.0 | 3.1 | 69.6 | 0.4 | 89.9 | 54.7 |
| Omeg. − {M} | 49.1 | 83.6 | 62.7 | 2.3 | 27.5 | 0.1 | 1.0 | 6.9 | 0.0 | 0.8 | 17.2 |
| **OmegAMP** | **56.9** | **90.4** | 43.5 | **0.3** | **138.1** | **0.0** | **0.4** | **0.5** | **0.0** | **0.4** | **0.7** |

## J.3 SENSITIVITY TO MISLABELLING

To evaluate the robustness of the OmegAMP classifier to potential label noise in the training data—specifically the risk that synthetic mutations might inadvertently produce active peptides—we conducted a sensitivity analysis. We performed 5-fold cross-validation where we systematically corrupted the training labels by flipping $M$ positive labels (EV AMPs) to negatives, simulating mislabeling events.

As illustrated in Fig. 7, the classifier's performance remains highly stable even as the number of corrupted labels increases. The AUPRC and Precision@100 metrics show negligible degradation until approximately 1,000 positive samples are corrupted (representing roughly one-third of the positive class). This resilience is attributable to our weighted loss function (see App. D.3), which prioritizes high-confidence EV data, allowing the model to learn robust decision boundaries despite the presence of noise.

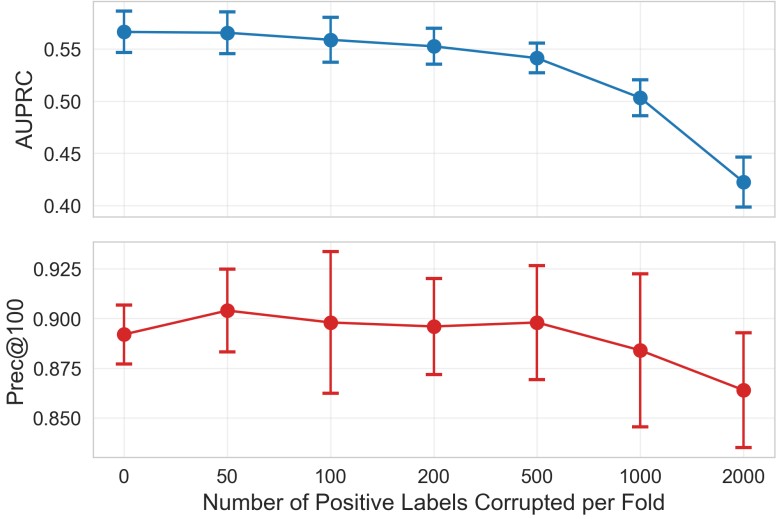

Figure 7: Robustness analysis of the OmegAMP classifier against training label noise. The plots track the model's performance in terms of AUPRC (top) and Precision@100 (bottom) as a function of the number of positive labels intentionally corrupted (flipped) per fold during training.

## J.4 EXTERNAL DATASET

To validate OmegAMP and contextualize its performance externally, we use the dataset from Tucker et al. (2018) to assess our classifier against all baselines reported in Sec. 4.1. This dataset contains random sequences, out of which less than 5% were shown to be AMPs. Importantly, our dataset and the aforementioned external dataset have distinct AMP definitions: our MIC-based criterion requires activity at $32\,\mu\text{g/mL}$, see App. F, contrasting with the more lenient definition in Tucker et al. (2018), which employs a surface display system to identify peptides with any detectable antimicrobial activity. Nevertheless, the negative examples from this dataset should be classified as inactive under

any definition. Therefore, a good classification scheme should be able to perform well, especially with respect to metrics that assess the TPR/FPR trade-off, like LR+.

Table 17: AMP classifiers performance on the dataset published by Tucker et al. (2018). Includes True Positive Rate (TPR), False Positive Rate (FPR), Positive Likelihood Ratio (LR+), overall Precision, and Precision@100.

|  | TPR | FPR | LR+ | Precision | Precision@100 |
|---|---|---|---|---|---|
| amPEPpy | 36.860 | 34.623 | 1.065 | 1.896 | 3.000 |
| AMPlify | 25.516 | 22.515 | 1.133 | 2.011 | 5.000 |
| AMPpredMFA | **100.000** | 100.000 | 1.000 | 1.779 | 2.000 |
| AMPScanner | 32.750 | 26.866 | 1.219 | 2.210 | 1.000 |
| HydrAMP-AMP | 34.864 | 31.452 | 1.109 | 1.972 | 4.000 |
| HydrAMP-MIC | 4.670 | 2.081 | 2.244 | 3.914 | 10.000 |
| SenseXAMP-classifier | 46.870 | 44.680 | 1.049 | 1.866 | **13.000** |
| PyAMPA | 2.265 | 1.409 | 1.607 | 2.833 | 5.000 |
| OmegAMP | 0.064 | **0.010** | **6.560** | **10.638** | 11.000 |

The results, presented in Tab. 17, indicate that most of the baseline methods greatly overestimate the number of positive samples in this dataset, as indicated by high FPRs. Additionally, all models but AMPpredMFA display TPR and FPR that are, on average, lower than those observed in our primary study, see Tab. 1. The aforementioned deviation between our experimentally validated dataset and the one from Tucker et al. (2018) is also reflected in this result. Notably, and consistent with our previous analysis, the LR+ and Prec@100 values showed other baselines performing near random-choice levels, while OmegAMP achieved statistically significant results. These findings from an independent test set further underscore OmegAMP's superior ability to discriminate between AMPs and non-AMPs, even across varying definitions.

### J.5 CLASSIFIER INTERPRETABILITY ANALYSIS

To investigate the biological principles learned by the OmegAMP general classifier, we conducted a feature importance analysis on the trained XGBoost model. The results show that a single feature, the mean charge of the peptide at physiological pH (7.0), was overwhelmingly dominant. This feature's high importance is underscored by its presence in 30% of the decision nodes across the tree ensemble. This finding confirms that the classifier has learned a well-established biological principle: the net positive charge of an AMP is a powerful predictor of its antimicrobial function. This charge facilitates the critical initial electrostatic attraction between the peptide and the negatively charged bacterial membranes, a key step for the peptide to exert its antimicrobial effect (Lei et al., 2019).

## K EXPERIMENTAL VALIDATION

To validate OmegAMP's design in real-world applications, we conducted rigorous experimental validation of OmegAMP-designed peptides. The primary target was to assess whether our framework's *in silico* performance translates into actual wet-lab biological activity. To this end, we synthesized 25 promising candidates and evaluated their efficacy against a panel of 17 clinically relevant bacterial strains, including eight multidrug-resistant (MDR) pathogens, which pose a significant threat to global health.

### K.1 EXPERIMENTAL DESIGN AND METHODS

**Peptide Design and Selection**   The 25 peptides for wet-lab validation were chosen through a comprehensive *in silico* pipeline designed to maximize the likelihood of success. We first generated two large pools of 50,000 candidate peptides using OmegAMP-SC (Subset Conditioning) and OmegAMP-PC (Property Conditioning). These pools were then filtered to retain only peptides with physicochemical properties within expert-defined ranges known to favor synthesizability and activity (charge: 2 to 10; length: 10 to 30; hydrophobicity: -0.5 to 0.8). After removing duplicates and sequences already present in known AMP databases, the remaining candidates were ranked using the OmegAMP general classifier and other predictors. Based on this final ranking, we selected 25 peptides for synthesis: 15 from the OmegAMP-SC pool and 10 from the OmegAMP-PC pool. The selected sequences are labeled s1, s2, ..., s25 to protect proprietary information.

Table 18: The 17 bacterial strains used for experimental validation. Strains marked with MDR are multi-drug resistant.

| ID | Bacterial Strain |
|---|---|
| AB1 | A. baumannii ATCC 19606 |
| AB2$_{MDR}$ | A. baumannii ATCC BAA-1605 |
| EC1 | E. coli ATCC 11775 |
| EC2 | E. coli AIC221 |
| EC3$_{MDR}$ | E. coli AIC222 |
| EC4$_{MDR}$ | E. coli ATCC BAA-3170 |
| KP1 | K. pneumoniae ATCC 13883 |
| KP2$_{MDR}$ | K. pneumoniae ATCC BAA-2342 |
| PA1 | P. aeruginosa PAO1 |
| PA2 | P. aeruginosa PA14 |
| PA3$_{MDR}$ | P. aeruginosa ATCC BAA-3197 |
| SE1 | S. enterica ATCC 9150 |
| SE2 | S. enterica Typhimurium ATCC 700720 |
| SA1 | S. aureus ATCC 12600 |
| SA2$_{MDR}$ | S. aureus ATCC BAA-1556 |
| EFS1$_{MDR}$ | E. faecalis ATCC 700802 |
| EFU1$_{MDR}$ | E. faecium ATCC 700221 |

**Bacterial Strains and Culture Conditions**  The pathogenic strains used in this study included *Acinetobacter baumannii* ATCC 19606; *A. baumannii* ATCC BAA-1605 (resistant to ceftazidime, gentamicin, ticarcillin, piperacillin, aztreonam, cefepime, ciprofloxacin, imipenem, and meropenem); *Escherichia coli* ATCC 11775; *E. coli* AIC221; *E. coli* AIC222 (resistant to polymyxin); *E. coli* ATCC BAA-3170 (resistant to colistin and polymyxin B); *Klebsiella pneumoniae* ATCC 13883; *K. pneumoniae* ATCC BAA-2342 (resistant to ertapenem and imipenem); *Pseudomonas aeruginosa* PAO1; *P. aeruginosa* PA14; *P. aeruginosa* ATCC BAA-3197 (resistant to fluoroquinolones, $\beta$-lactams, and carbapenems); *Salmonella enterica* ATCC 9150; *S. enterica* subsp. *enterica* Typhimurium ATCC 700720; *Staphylococcus aureus* ATCC 12600; *S. aureus* ATCC BAA-1556 (methicillin-resistant); *Enterococcus faecalis* ATCC 700802 (vancomycin-resistant); and *Enterococcus faecium* ATCC 700221 (vancomycin-resistant). Tab. 18 presents a structured view of these strains. Additionally, Pseudomonas strains were grown on selective Pseudomonas Isolation Agar, whereas all other bacteria were cultured in Luria-Bertani (LB) agar and broth. Each strain was initiated from a single colony, incubated overnight at $37\,°\mathrm{C}$, and then diluted 1:100 into fresh medium to reach mid-logarithmic growth phase.

**Minimal Inhibitory Concentration (MIC) Assays**  MIC values were determined by broth microdilution using untreated 96-well microplates. Peptides were prepared as twofold serial dilutions (1-64 $\mu\mathrm{mol\,L^{-1}}$) in sterile water and mixed at a 1:1 ratio with LB medium containing $4\times10^6\,\mathrm{CFU\,mL^{-1}}$ of bacteria. The MIC was defined as the lowest peptide concentration that fully prevented visible bacterial growth after 24 h of incubation at $37\,°\mathrm{C}$. Each assay was performed independently in triplicate.

K.2 OMEGAMP PEPTIDES SHOW HIGH POTENCY AND VALIDATE CLASSIFIER DESIGN

**Unprecedented Experimental Hit Rate, Broad-Spectrum Activity and High Potency**  We share the measured MIC values (the lower, the better, with MIC $\leq 32\mu$g/mL considered the activity threshold) for all considered strains in Tab. 19. A remarkable 24/25 sequences are active against at least one bacterial strain, respectively. Therefore, yielding a 24/25=96% hit rate, which, to the best of our knowledge, is the highest reported hit rate for any AMP experiment. Additionally, two of the peptides (s2 and s23) are active against all tested strains, and overall, the peptides show broad-spectrum activity.

Table 19: Experimentally measured Minimum Inhibitory Concentration (MIC, in $\mu$g/mL) for 25 OmegAMP-generated peptides against Gram-negative and Gram-positive bacterial strains. '-' indicates no observed activity.

| ID | AB1 | AB2 | EC1 | EC2 | EC3 | EC4 | KP1 | KP2 | PA1 | PA2 | PA3 | SE1 | SE2 | SA1 | SA2 | EFS1 | EFU1 |
|-----|-----|-----|-----|-----|-----|-----|-----|-----|-----|-----|-----|-----|-----|-----|-----|------|------|
| s1 | 4 | 8 | 32 | 8 | 16 | 32 | 16 | 64 | - | 32 | - | 8 | 16 | 64 | 32 | - | 1 |
| s2 | 2 | 2 | 8 | 2 | 2 | 2 | 4 | 2 | 16 | 2 | 8 | 1 | 2 | 16 | 16 | 16 | 2 |
| s3 | 4 | 4 | 4 | 4 | 2 | 4 | 8 | 16 | 64 | 16 | 32 | 4 | 8 | - | - | - | 8 |
| s4 | 8 | 8 | 32 | 4 | 16 | 8 | - | - | 16 | 16 | 8 | 2 | 8 | 32 | 64 | - | 16 |
| s5 | 16 | 32 | - | - | - | - | - | - | - | - | - | - | 16 | 4 | - | - | 8 |
| s6 | 8 | 4 | 64 | 4 | 16 | 16 | 64 | - | 16 | 16 | 16 | 4 | 8 | 64 | - | - | 16 |
| s7 | 2 | 4 | 8 | 2 | 8 | 4 | 64 | 8 | 4 | 8 | 2 | 1 | 8 | 32 | 64 | - | 4 |
| s8 | 4 | 8 | 32 | 8 | 16 | 32 | 16 | - | 32 | - | 32 | 2 | 16 | 64 | - | - | 2 |
| s9 | 4 | 2 | 32 | 8 | 16 | 4 | - | - | 64 | 16 | 32 | 4 | 8 | - | - | - | 8 |
| s10 | 2 | 2 | 32 | 4 | 4 | 4 | 32 | 8 | 32 | 32 | 8 | 1 | 4 | 16 | 32 | - | 4 |
| s11 | 32 | 16 | 32 | 16 | 8 | 16 | 32 | - | - | 64 | - | 32 | 32 | 16 | 16 | 64 | 16 |
| s12 | - | - | - | - | - | - | - | - | - | - | - | - | - | 64 | - | - | - |
| s13 | 32 | 32 | 8 | 8 | 16 | 8 | - | - | - | 16 | 64 | 8 | 32 | 64 | - | 32 | 4 |
| s14 | - | - | - | - | - | - | - | - | - | - | - | - | - | 4 | - | - | 32 |
| s15 | 4 | 16 | - | 16 | 16 | 8 | - | - | - | - | - | 16 | 2 | 1 | - | 32 | 1 |
| s16 | 4 | 2 | 4 | 2 | 2 | 4 | 1 | 4 | 8 | 2 | 4 | 2 | 2 | 16 | 32 | - | 2 |
| s17 | 16 | 4 | 32 | 4 | 16 | 8 | - | - | 8 | 16 | 1 | 1 | 4 | - | 32 | - | 1 |
| s18 | 8 | 4 | - | 32 | - | 8 | - | - | - | - | - | 8 | 32 | - | - | - | 4 |
| s19 | 4 | 2 | 16 | 2 | 2 | 2 | - | - | 4 | 4 | 2 | 1 | 4 | - | 64 | - | 2 |
| s20 | 16 | 8 | 16 | 4 | 16 | 8 | 64 | 32 | 4 | 8 | 2 | 2 | 4 | 32 | 32 | - | 1 |
| s21 | 8 | 8 | 32 | 4 | 16 | 8 | - | 32 | 32 | 16 | 16 | 2 | 8 | - | - | - | 2 |
| s22 | 1 | 4 | 8 | 4 | 8 | 4 | 2 | 8 | 16 | 8 | 8 | 2 | 4 | - | - | - | 2 |
| s23 | 2 | 2 | 4 | 1 | 2 | 2 | 16 | 8 | 2 | 2 | 1 | 1 | 2 | 8 | 8 | 8 | 1 |
| s24 | 16 | 4 | 64 | 16 | 32 | 4 | - | - | 32 | 32 | 4 | 2 | 8 | 64 | - | - | 2 |
| s25 | 64 | 16 | - | 32 | 32 | 16 | - | - | - | 64 | 2 | 2 | 16 | - | - | - | 8 |

The clinical relevance and high potency of the generated peptides are further underscored by their per-strain hit rates at stringent MIC thresholds (Tab. 20). The analysis reveals exceptional performance: for every bacterial strain tested, at least one peptide demonstrated high potency with an MIC of $\leq 8\mu$g/mL, as observed by the non-zero values for all values in the $\leq 8$ column. Crucially, this effectiveness extends to the most challenging bacteria, with numerous peptides showing strong activity against multi-drug resistant strains (AB2, EC3, EC4, KP2, PA3, SA2, EFS1, EFU1). These results confirm OmegAMP's ability to generate potent and therapeutically relevant candidates for high-priority pathogens.

Table 20: Fraction of the 25 tested peptides active against each bacterial strain at various MIC thresholds ($\mu$g/mL).

| ID | $\leq 1$ | $\leq 2$ | $\leq 4$ | $\leq 8$ | $\leq 16$ | $\leq 32$ | $\leq 64$ |
|-----|------|------|------|------|------|------|------|
| AB1 | 0.04 | 0.20 | 0.48 | 0.64 | 0.80 | 0.88 | 0.92 |
| AB2$_{MDR}$ | 0.00 | 0.24 | 0.52 | 0.72 | 0.84 | 0.92 | 0.92 |
| EC1 | 0.00 | 0.00 | 0.12 | 0.28 | 0.36 | 0.68 | 0.76 |
| EC2 | 0.04 | 0.20 | 0.52 | 0.68 | 0.80 | 0.88 | 0.88 |
| EC3$_{MDR}$ | 0.00 | 0.20 | 0.24 | 0.36 | 0.76 | 0.84 | 0.84 |
| EC4$_{MDR}$ | 0.00 | 0.12 | 0.40 | 0.68 | 0.80 | 0.88 | 0.88 |
| KP1 | 0.04 | 0.08 | 0.12 | 0.16 | 0.28 | 0.36 | 0.48 |
| KP2$_{MDR}$ | 0.00 | 0.04 | 0.08 | 0.24 | 0.28 | 0.36 | 0.40 |
| PA1 | 0.00 | 0.04 | 0.16 | 0.24 | 0.40 | 0.56 | 0.64 |
| PA2 | 0.00 | 0.12 | 0.16 | 0.28 | 0.56 | 0.68 | 0.76 |
| PA3$_{MDR}$ | 0.08 | 0.24 | 0.32 | 0.48 | 0.60 | 0.72 | 0.76 |
| SE1 | 0.24 | 0.60 | 0.72 | 0.84 | 0.88 | 0.92 | 0.92 |
| SE2 | 0.00 | 0.12 | 0.32 | 0.60 | 0.72 | 0.84 | 0.84 |
| SA1 | 0.00 | 0.00 | 0.00 | 0.04 | 0.20 | 0.32 | 0.52 |
| SA2$_{MDR}$ | 0.00 | 0.00 | 0.00 | 0.04 | 0.12 | 0.36 | 0.48 |
| EFS1$_{MDR}$ | 0.00 | 0.00 | 0.00 | 0.04 | 0.08 | 0.12 | 0.16 |
| EFU1$_{MDR}$ | 0.20 | 0.48 | 0.64 | 0.80 | 0.92 | 0.96 | 0.96 |

**Classifier Backtest Validates Low FPR Goal** Following the wet-lab experiments, we performed a backtest analysis to evaluate how well our classifiers' *in silico* predictions held up against the experimental ground truth. The results, shown in Tab. 21, strongly validate our design philosophy of prioritizing high specificity to minimize

Table 21: Classifier performance metrics of the 25 experimentally evaluated peptides.

| Classifier | P | TP | FP | TN | FN | TPR | FPR |
|---|---|---|---|---|---|---|---|
| General | 24 | 11 | 0 | 1 | 13 | 0.46 | 0.00 |
| Species - *A. baumannii* | 23 | 8 | 0 | 2 | 15 | 0.35 | 0.00 |
| Species - *E. coli* | 22 | 8 | 0 | 3 | 14 | 0.36 | 0.00 |
| Species - *K. pneumoniae* | 12 | 3 | 1 | 12 | 9 | 0.25 | 0.08 |
| Species - *P. aeruginosa* | 20 | 9 | 0 | 5 | 11 | 0.45 | 0.00 |
| Species - *S. aureus* | 11 | 5 | 2 | 12 | 6 | 0.45 | 0.14 |

costly false positives. Across the general and most species-specific models, the False Positive Rate (FPR) was exceptionally low, reaching 0.00 for four of the six classifiers. This confirms that the models are highly effective at correctly identifying inactive peptides — a critical objective for reducing experimental costs. The reported True Positive Rate (TPR) is consistent with our *in silico* findings, see Sec. 4.1, showing that the observed *in silico* performance translates to real-world settings. Ultimately, these findings show that OmegAMP classifiers serve as a stringent, high-confidence filter, ensuring that peptides selected for synthesis have a high probability of being active, thereby enhancing the overall efficiency and success rate of the discovery pipeline.

## L   LIMITATIONS & FUTURE WORK

**Contextual Specificity of Activity Definition.**   The definition of peptide activity, as detailed in App. F, is inherently linked to the specific bacterial strains incorporated into the analysis. This strain-dependent characterization is a well-understood consideration within the broader field of antimicrobial peptide research, rather than a limitation unique to our model or approach. It follows that a peptide active against a strain not included in our study would be classified as inactive under our defined criteria, irrespective of its potential efficacy against a wider spectrum of bacteria. Consequently, our reported results and conclusions regarding peptide activity are interpreted within the specific context of the evaluated strains, a necessary and common delimitation when investigating the complex landscape of peptide-bacterial interactions.

**Generalization to Novel Peptide Space and Property Extrapolation.**   While our experiments in Sec. 4 demonstrate robust performance for both generative and discriminative models on important benchmarks, their application to truly novel chemical and functional spaces presents considerations common to many data-driven models. Current publicly available peptide datasets, upon which our models are trained, often exhibit certain prevalent structural and activity patterns. Consequently, assessing performance on peptides with radically divergent characteristics, or those representing entirely new classes of activity, remains an open and important challenge for the field. Similarly, our evaluation of property conditioning focused on target values within the empirically observed ranges of the training data, where performance is well-characterized. The ability of conditional generative models to reliably extrapolate to property combinations significantly outside these validated ranges is an active area of research, and performance in such regimes warrants dedicated future investigation.

# M   REBUTTAL OUTCOMES

**Extending the Conditioning Space**   A key advantage of OmegAMP is the flexibility of its conditioning mechanism. To demonstrate the ability to incorporate additional biophysical properties, we retrained the model with the *Instability Index* included in the conditioning vector. This property serves as a proxy for protease stability.

Results in Tab. 22 show that the model successfully incorporates this new constraint. While there is a trade-off, as controlling the Instability Index is inherently more challenging (MAE 0.86) and leads to a slight reduction in control over other properties, the model generates peptides with higher overall Fitness Scores (0.14). This confirms that OmegAMP can be extended to target multi-objective criteria including stability, toxicity, or other computable properties, provided reliable data exist.

Table 22: Ablation study on extending the conditioning space. We compare the standard OmegAMP with a version trained to also control the Instability Index.

| Gen. Model | Omeg. Class. | Fit. Score | Div. | Uniq. | MAE (in std units) | | | |
|---|---|---|---|---|---|---|---|---|
| | | | | | Length | Charge | Hydroph. | Inst. Index |
| OmegAMP w/ Inst. Index | 8.7 | **0.14** | **0.64** | **96** | 0.19 | 0.17 | 0.20 | 0.86 |
| OmegAMP | **10.5** | 0.13 | **0.64** | 94 | **0.04** | **0.16** | **0.18** | - |

**Generative Evaluation with External Classifiers**   To ensure that the high predicted activity of OmegAMP-generated sequences is not an artifact of system-internal consistency bias (i.e., scoring samples with a classifier trained on similar data distributions), we evaluated the generative models using two independent, third-party classifiers: amPEPpy and AMPlify. These were selected for their strong baseline performance in Tab. 1.

The results in Tab. 23 confirm the findings reported in the main text. OmegAMP variants consistently achieve the highest predicted activity rates across all external classifiers. Notably, OmegAMP-PC achieves 87.7% and 84.5% predicted positives on amPEPpy and AMPlify, respectively, significantly outperforming baseline generative models like AMPGAN and Diff-AMP. This cross-validation by independent models provides strong evidence that OmegAMP generates high-quality peptide candidates with genuine antimicrobial potential.

Table 23: Performance comparison across generative models using independent external classifiers (amPEPpy, AMPlify) to verify generated sequence quality.

| Gen. Model | HydrAMP-MIC | OmegAMP Class. | amPEPpy | AMPlify | Fitness Score | Diversity | Uniqueness | Novelty |
|---|---|---|---|---|---|---|---|---|
| EV AMPs (*Data*) | 81.6 | 43.5* | 94.1 | 96.1 | 0.16 | 0.62 | - | - |
| AMPGAN | 31.6 | 0.3 | 50.4 | 49.1 | 0.10 | 0.57 | **100** | **100** |
| Diff-AMP | 27.8 | 0.0 | 50.4 | 13.5 | 0.08 | 0.63 | **100** | **100** |
| HydrAMP | 44.1 | 0.0 | 56.4 | 59.3 | 0.09 | **0.70** | **100** | **100** |
| AMP-Diffusion | 42.8 | 2.2 | 28.9 | 11.8 | 0.11 | 0.64 | 91 | **100** |
| OmegAMP | 33.8 | 10.5 | 65.3 | 64.1 | 0.13 | 0.64 | 94 | 98 |
| OmegAMP-PC | **70.2** | 14.8 | **87.7** | 84.5 | **0.16** | 0.60 | 98 | 99 |
| OmegAMP-SC | 64.1 | **16.4** | 87.1 | **88.9** | 0.15 | 0.61 | 95 | 97 |

