# OpenReview forum: "OmegAMP: Targeted AMP Discovery via Biologically Informed Generation"
_ICLR.cc/2026/Conference — Submitted to ICLR 2026_

### Official Review · Reviewer_CTVM · 2025-10-26

**Soundness:** 3
**Presentation:** 3
**Contribution:** 2
**Rating:** 4
**Confidence:** 3

**Summary:**

The paper introduces OmegAMP, a diffusion-based framework for controllable antimicrobial peptide (AMP) generation that couples a biologically informed embedding with a flexible conditioning scheme over AMP label, length, charge, and hydrophobicity.

It proposes two targeting modes Property Conditioning (ranges) and Subset Conditioning (derived from exemplar sequences)—to steer generation, including toward species-specific activity profiles.

The authors further design a stringent AMP filtering stage using XGBoost classifiers trained with a weighted loss and synthetic negative augmentation (random, shuffled, mutated), drastically reducing false positives on challenging negatives (signal, metabolic, added-deleted).

Extensive in silico evaluations show SOTA generation and classification metrics, and wet-lab validation on 25 peptides yields a 96% hit rate with high potency, including efficacy against multi-drug resistant strains.

**Strengths:**

Originality: The biologically informed, invertible residue embedding tied to a diffusion model plus the dual conditioning strategy (property/subset) is a novel and practical way to achieve fine-grained and species-targeted controllability. The synthetic-negative training and weighted loss for classifiers directly address the chronic high-FPR problem in AMP screening.

Quality: The work presents thorough evaluations: ablations on embedding scales, conditional control MAEs, classifier robustness on multiple hard negative sets, external dataset testing, and comprehensive wet-lab validation with MIC distributions across 17 strains. Clear reporting of metrics (AUPRC, Prec@100, LR+, diversity/uniqueness/novelty, fitness) and careful cross-validation support the claims.

Clarity and significance: The paper is well-structured with precise methodological detail (conditioning objective, CADS, embedding injectivity/decoding, classifier features/loss), clear figures/tables, and actionable guidance for practitioners. Demonstrating near-perfect wet-lab hit rates and strong potency against MDR pathogens indicates high potential impact for antimicrobial discovery.

**Weaknesses:**

Conditioning expressivity: The conditional-based generation is not a new idea (Accelerated antimicrobial discovery via deep generative models and molecular dynamics simulations). Besides, the current cond(·) vector captures AMP label, length, charge, and hydrophobicity, but omits other relevant biophysical/ADMET factors (e.g., protease stability, hemolysis, toxicity, aggregation, secondary structure motifs). Extending the conditioning space and validating multi-objective trade-offs would increase practical utility.

Dataset and bias considerations: Generative training uses AMP-like general peptides from Peptipedia and EV labels from DBAASP with standardization filters, which may introduce selection bias and label noise. More analysis on data quality, potential leakage, and sensitivity to dataset composition (e.g., species distribution) would strengthen reproducibility and fairness claims.

**Questions:**

Could the conditioning vector be expanded to include toxicity/hemolysis proxies or structural targets (e.g., predicted helicity, aggregation propensity), and how would that affect classifier precision and wet-lab hit rates?

For species-specific generation, what is the sensitivity of Subset Conditioning to the size/quality of the exemplar set, and can you provide guidance on the minimal number and diversity of exemplars needed to achieve gains?

---

> ### Author Response · Authors · 2025-11-21
> **Part 1/2**
>
> We thank the reviewer for their thoughtful feedback. We appreciate that reviewer CVTM acknowledges that OmegAMP’s ability to design peptides active against drug-resistant pathogens, as established with our wet lab validation, indicates high potential impact for antimicrobial discovery. At a more granular level, we are encouraged that the reviewer finds our biologically-inspired embedding scheme, our conditioning strategies, and our stringent classifiers convincing.
>
> &nbsp;
>
> ### 1. Rationale for Conditioning Strategy and Extension to Instability Index
>
> > Conditioning expressivity: The conditional-based generation is not a new idea [...] omits other relevant biophysical/ADMET factors [...] Extending the conditioning space and validating multi-objective trade-offs would increase practical utility.
> > Could the conditioning vector be expanded to include toxicity/hemolysis proxies or structural targets [...]?
>
> We thank the reviewer for their observation. It is true that conditional-based generation is not a new idea, however, previous approaches suffer from key limitations:
> *   **Reliance on Inferred Labels:** Existing methods, like the one suggested by the reviewer [1], typically condition on binary functional attributes (e.g., activity, toxicity). Scaling this approach with limited data often involves training external classifiers to infer these labels.
> *   **Noise Propagation:** Such classifiers introduce a fundamental bottleneck: since AMP classifiers often exhibit significant error rates, the generative model is inevitably trained on noisy estimates rather than ground truth.
> *   **Lack of Generalizability:** Reliance on specific binary tags (e.g., activity against strain X) severely limits generalizability, as such models cannot inherently extend to species or strains unseen during training.
> *   **Limited Physicochemical Control:** Focusing solely on predicted activity limits the ability to control physicochemical properties critical for **synthesis success**, such as hydrophobicity and charge (see section 4.3 in L452-468).
>
> Our approach sidesteps these limitations:
> *   **Noise-Free Labels:** We condition on properties that are deterministically computable for any sequence, ensuring the model is trained on exact labels rather than noisy classifier estimates.
> *   **Synthesis Relevance:** We prioritize length, charge, and hydrophobicity, as these are critical for both biological function and synthesis success.
> *   **Inherent Generalizability:** By avoiding species-specific tags in favor of universal physicochemical properties, our framework can be used to tackle unseen species (via Subset Conditioning/Property Conditioning).
>
> Furthermore, regarding the inclusion of additional properties like toxicity or structural motifs:
> *   **Model Extensibility:** Extending the conditioning space of our model is perfectly plausible, provided reliable training labels exist. We excluded toxicity/ADMET only due to the lack of reliable ground truth data [2].
> *   **Proof of Flexibility (Instability Index):** To demonstrate this capability, we re-trained our model adding the **Instability Index**. This feature is a computable proxy for protease stability, as requested by the reviewer, and can be readily obtained for every peptide sequence.
>
> The results of this experiment suggest:
> *   **Generative Trade-offs:** We observe mixed conclusions in terms of realistic generation, as the fraction of predicted AMPs decreased but the overall Fitness Score increased.
> *   **Controllability Impact:** Adding the Instability Index conditioning feature slightly decreased the controllability of the other conditioning features.
> *   **Optimization Challenge:** Controlling the Instability Index seems to be much more challenging than the other features, as the MAE-by-std is significantly higher.
>
> |**Gen. Model**|**OmegAMP Class.**|**Fit. Score**|**Div.**|**Uniq.**|**MAE-by-std (Length)**|**MAE-by-std (Charge)**|**MAE-by-std (Hydroph.)**|**MAE-by-std (Inst. Index)**|
> |:---|:---:|:---:|:---:|:---:|:---:|:---:|:---:|:---:|
> |OmegAMP w/ Inst. Index|8.7|**0.14**|**0.64**|**96**|0.19|0.17|0.20|0.86|
> |OmegAMP|**10.5**|0.13|**0.64**|94|**0.04**|**0.16**|**0.18**|-|
>
> We updated L162-164 and Appendix M (L1728-1768) with these insights.
>
> &nbsp;

---

> > ### Author Response · Authors · 2025-11-21
> > **Part 2/2**
> >
> > ### 2. Addressing Dataset Bias and Label Noise
> >
> > > Dataset and bias considerations: [...] may introduce selection bias and label noise. More analysis on data quality, potential leakage, and sensitivity to dataset composition [...] would strengthen reproducibility and fairness claims.
> >
> > We agree with the reviewer on the importance of dataset selection. Current publicly available peptide datasets, upon which our models are trained, often exhibit certain prevalent structural and activity patterns, as detailed in Appendix L (L1691-1712). In addition, we would like to highlight that:
> > *   **Methodological Consistency:** Our dataset selection is consistent with existing methods [3].
> > *   **Database Curation:** Despite being imperfect, these databases are highly curated [4].
> >
> > Additionally, we take appropriate measures to mitigate these risks:
> > * **Dataset Construction:** EV labels were curated to standardize the experimental conditions and mitigate any technical noise.
> > * **Filtering:** The classifier is trained with a weighted loss function that prioritizes the contribution of EV labels.
> > * **Generation:** The conditioning vector takes into account the data source through the “isAMP” property.
> >
> > Moreover, we perform experiments that directly address the reviewers’ requests:
> > * **Sensitivity to Dataset Composition:** We retrain our generative model without the AMP-like general peptides from Peptipedia, and show that the inclusion of these general peptides improves generative performance, see Appendix M (L1728-1768).
> > * **Data Quality:** We measure the impact of mislabelling events to classifier training, and conclude that our classifier training optimization is robust to mislabelled datapoints, see Appendix J.3 (L1469-1503).
> >
> > Both of these experiments are explored in detail in sections 2. and 3. of our response to reviewer ieQ4.
> >
> >
> > &nbsp;
> >
> > ### 3. Guidelines for Subset Conditioning: Quality vs. Quantity
> >
> > > [...] what is the sensitivity of Subset Conditioning to the size/quality of the exemplar set, and can you provide guidance on the minimal number and diversity of exemplars needed to achieve gains?
> >
> > Thank you for the insightful question. Subset Conditioning (SC) works by conditioning the generative model to design peptides whose joint property distribution reflects that of the exemplar set. Therefore, performance is far more sensitive to the **quality** of the exemplars than to the **size** of the set.
> >
> > *   **Sensitivity (Quality vs. Size):** The key factor is how strongly the properties of the exemplars correlate with antimicrobial activity. A small, carefully curated set of highly potent peptides effectively guides generation toward high-activity regions. Conversely, a large set of weak or moderate peptides will instruct the model to generate candidates with similarly mediocre properties.
> > *   **Empirical Validation:** We empirically demonstrate this in Figure 2a (L378-390), where SC conditioned on  smaller, species-specific sets (e.g. <1000 sequences) outperforms larger, general EV set (~4,000 sequences) when targeting specific species like E. coli. The larger set, despite being more numerous, contained peptides not verified against the specific target strain, confirming that increasing dataset size cannot compensate for lower specificity and potency.
> > *   **Guidance on Quantity:** There is no fixed minimum number required. A single, highly effective exemplar is sufficient to improve hit rates if its physicochemical properties are strongly indicative of activity.
> > *   **Guidance on Diversity:** We recommend using a diverse set of high-quality exemplars. A diverse set incentivizes the model to replicate a broader distribution of successful property combinations, leading to a more varied pool of candidate peptides.
> >
> > In summary, our guidance is to prioritize a **high-quality and diverse** exemplar set. The focus should be on selecting peptides with confirmed, potent activity against the target species, rather than simply maximizing the number of exemplars. We updated our manuscript to reflect these insights (L431-435).
> >
> > &nbsp;
> > We sincerely thank Reviewer CVTM for their evaluation, specifically their recognition of our wet-lab success and methodological rigor. The feedback regarding conditioning flexibility and dataset sensitivity has helped us demonstrate the adaptability of OmegAMP. We look forward to further discussion.
> >
> > ---
> >
> > [1] Das, P., et al. Accelerated antimicrobial discovery via deep generative models and molecular dynamics simulations. *Nature Biomedical Engineering*
> > [2] Szymczak, P., et al. Artificial intelligence-driven antimicrobial peptide discovery. *Current Opinion in Structural Biology*
> >
> > [3] Szymczak, P., et al. Discovering highly potent antimicrobial peptides with deep generative model HydrAMP. *Nature Communications*
> >
> > [4] Pirtskhalava, M., et al. DBAASP v3: database of antimicrobial/cytotoxic activity and structure of peptides as a resource for development of new therapeutics. *Nucleic acids research*

---

### Official Review · Reviewer_ieQ4 · 2025-10-31

**Soundness:** 3
**Presentation:** 3
**Contribution:** 2
**Rating:** 4
**Confidence:** 3

**Summary:**

The paper proposes an integrated framework, OmegAMP, for antimicrobial peptide (AMP) discovery, comprising: diffusion-based sequence generation (with biologically inspired residue-level embeddings and flexible conditional control) and low–false-positive filtering classifiers (XGBoost with synthetic negative data augmentation and a weighted loss).

**Strengths:**

The paper proposes an integrated framework, OmegAMP, for antimicrobial peptide (AMP) discovery, comprising: diffusion-based sequence generation (with biologically inspired residue-level embeddings and flexible conditional control) and low–false-positive filtering classifiers (XGBoost with synthetic negative data augmentation and a weighted loss).

**Weaknesses:**

1. Scoring generated samples with the authors’ own classifier inevitably introduces system-internal consistency bias (despite providing HydrAMP-MIC as a secondary scorer). It is recommended to include more external independent classifiers or larger-scale blinded experimental test sets to reduce evaluator–generator coupling.
2. While “random/shuffled” negatives are reasonable, introducing only five point mutations does not necessarily render sequences inactive, leading to potential mislabeling. Although the paper uses a weighted loss to down-weight non-EV data, it is advisable to quantitatively report sensitivity of performance to mislabeling rates, or increase mutation strength/proportion as a control, to more robustly define non-AMPs.
3. The generative model’s “large-scale general peptides” are derived from Peptipedia’s predicted label set (non-EV), which may be noisy. Although the authors acknowledge and distinguish this, they should more clearly quantify how such noise affects the generative distribution and controllability (e.g., a training ablation without that data).
4. The “conditioning vector” for subset conditioning includes only global attributes (length, net charge, hydrophobicity, AMP tag), without explicitly encoding species/strain information or sequence motif/positional patterns. If species selectivity relies on finer sequence–structure features, aligning global distributions alone may be insufficient.
5. The claimed species-specific “success” is mainly indirectly supported by elevated classifier scores, while feature analysis indicates “mean net charge” is highly dominant; this may cause species specificity to appear merely as matching certain global attribute distributions.

**Questions:**

Introduce third-party independent classifiers or conduct “blinded scoring” with collaborators to reduce system-internal bias.
Systematically study how different mutation ratios/patterns affect classifier FPR/TPR and report sensitivity curves, or align training with a larger EV non-AMP set to mitigate mislabeling risk.

---

> ### Author Response · Authors · 2025-11-21
> **Part 1/2**
>
> We thank Reviewer ieQ4 for their valuable feedback and questions. Since the review’s 'Strengths' section focuses on summarizing our method, we want to ensure our wet-lab contributions are fully considered. Our approach achieved a 96% hit rate (25 sequences confirmed) against multi-drug resistant strains, demonstrating practical real-world activity which is the main goal of any AMP generative approach, going beyond standard computational benchmarks.
>
>
> &nbsp;
>
> ### 1. Mitigating System-Internal Bias
>
> > Scoring samples with the authors’ own classifier inevitably introduces system-internal consistency bias […]
> > Introduce independent classifiers [...]
>
> We follow the reviewer’s suggestion to include more external independent classifiers:
> *   **External Validation:** We evaluated all generative models with **amPEPpy** and **AMPlify**, selected for their superior AUPRC and Prec@100 performance (see T1, L270-283).
> *   **Confirmation of Quality:** These external classifiers also identify OmegAMP’s generated sequences as AMPs, supporting that the sequence selection based on the OmegAMP classifier does not suffer from a consistency bias that drives the results meaningless.
>
>
> |**Gen. Model**|**HydrAMP-MIC**|**OmegAMP Class.**|**amPEPpy**|**AMPlify**|**Fitness Score**|**Diversity**|**Uniqueness**|**Novelty**|
> |:---|---:|---:|---:|---:|---:|---:|---:|---:|
> |EV AMPs (*Data*)|81.6|43.5*|94.1|96.1|0.16|0.62|-|-|
> |AMPGAN|31.6|0.3|50.4|49.1|0.10|0.57|**100**|**100**|
> |Diff-AMP|27.8|0.0|50.4|13.5|0.08|0.63|**100**|**100**|
> |HydrAMP|44.1|0.0|56.4|59.3|0.09|**0.70**|**100**|**100**|
> |AMP-Diffusion|42.8|2.2|28.9|11.8|0.11|0.64|91|**100**|
> |OmegAMP|33.8|10.5|65.3|64.1|0.13|0.64|94|98|
> |OmegAMP-PC|**70.2**|14.8|**87.7**|84.5|**0.16**|0.60|98|99|
> |OmegAMP-SC|64.1|**16.4**|87.1|**88.9**|0.15|0.61|95|97|
>
> We updated the paper accordingly in App.M (L1728-1768).
>
> Additionally, we would like to point out specific design choices from OmegAMP that directly mitigate consistency bias:
> *   **Model Independence:** The generative and classifier models are trained separately.
> *   **Data Separation:** The datasets differ significantly; the classifier includes 300k synthetic negatives, while the generative model uses ~800k sequences from Peptipedia. These sequences are not shared (see App.C, L877-943).
>
> **Ultimately, our wet-lab results constitute the strongest evidence against system-internal bias.** Since consistency bias artificially inflates scores for inactive artifacts, achieving a **96% hit rate (24/25 confirmed)** against MDR strains would be impossible if the classifiers were simply reinforcing the generator's internal distribution rather than capturing true biological constraints.
>
> &nbsp;
>
> ### 2. Sensitivity Analysis: Robustness to Label Noise
>
> > [...] introducing only 5 point mutations does not necessarily render sequences inactive, leading to potential mislabeling. [...] it is advisable to report sensitivity of performance to mislabeling rates [...]
>
> > Systematically study how different mutation ratios/patterns affect classifier FPR/TPR and report sensitivity curves [...]
>
> We appreciate the observation and acknowledge the importance of motivating the inactivity of mutated sequences.
> *   **Distributional Shift:** Fig. 4 (App.G, L1070-1171) shows notable shifts between AMPs and mutated sequences, particularly in pseudo-perplexity distributions.
> *   **Mutation Magnitude:** With a mean length of ~30 AA, we chose to mutate 5 positions as this represents ~17% of the sequence, which we expect to result in a significant structural deviation.
> *   **Classifier Separation:** The ability of classifiers to distinguish these sets (T.1, L270-283) implies that a meaningful decision boundary exists between these sets. Note that all reported results are based on completely unseen sequences.
> *   **Loss Weighting:** We substantially down-weight non-EV data, limiting the impact of potential mislabels.
>
> To quantitatively measure the impact of mislabeling:
> *   **Experiment:** We performed 5-fold cross-validation while randomly corrupting $M$ positive labels into negative ones to simulate mislabeling.
> *   **Result:** OmegAMP’s classifier shows excellent robustness. AUPRC and Prec@100 remain stable until >1000 events (corrupting ~1/3 of positive training data).
>
> For more additional details, we kindly refer the reviewer to App.J.3 (L1469-1499).
>
> &nbsp;
>
> ### 3. Ablation Study: The Role of General Peptide Sequences
>
> > The generative model’s “large-scale general peptides” [...] may be noisy. [...] more clearly quantify how such noise affects the generative distribution and controllability (ablate without that data).
>
> We followed the reviewer’s suggestion and performed an ablation training the generative model without “General Peptide Sequences”.
>
> *   **Result:** The ablation demonstrates that training with a large, diverse set of sequences yields a more controllable model that produces sequences with higher average antimicrobial activity.

---

> > ### Author Response · Authors · 2025-11-21
> > **Part 2/2**
> >
> > |**Gen. Model**|**Omeg. Class.**|**Fit. Score**|**Div.**|**Uniq.**|**MAE-by-std (Length)**|**MAE-by-std (Charge)**|**MAE-by-std (Hydroph.)**|
> > |:---|---:|---:|---:|---:|---:|---:|---:|
> > |OmegAMP w/o “General Peptide Sequences”|8.1|0.11|0.60|**96**|0.23|0.27|0.27|
> > |OmegAMP|**10.5**|**0.13**|**0.64**|94|**0.04**|**0.16**|**0.18**|
> >
> > We update the paper with these insights in App.I.1, L1236-1258.
> >
> > As the reviewer acknowledged in their feedback, we motivate the inclusion of the “General Peptide Sequences” in the following way:
> > *   **Data Diversity:** By using such a large dataset, we expose the model to a larger and chemically diverse set of sequences during training, improving overall generation quality and controllability.
> > *   **Learned Correlations:** The rationale is that the connection between physicochemical properties and amino acid sequence can also be learned from general peptides.
> > *   **Source Distinction:** We preserve the distinction between these two data sources to enable the generative model to isolate critical patterns related to verified antimicrobial activity.
> >
> > &nbsp;
> >
> > ### 4. Capturing Species Specificity via Global Attributes
> >
> > > The “conditioning vector” for SC includes only global attributes, without encoding species information or sequence patterns. [...] aligning global distributions alone may be insufficient.
> >
> > We agree with the reviewer that global attributes alone may not fully capture the nuances of species selectivity. Yet, we emphasize that:
> >
> > *   **Broad-Spectrum Activity:** AMPs often exhibit cross-species activity driven by global physicochemical properties (e.g., charge and hydrophobicity). Optimizing these global features helps target broad-spectrum efficacy and ensures the generated candidates occupy the correct physicochemical regions for biological function and synthesis success.
> > *   **Specificity via SC:** Even though the conditioning vector is global, our **Subset Conditioning** strategy implicitly captures finer distinctions. By sampling property vectors from peptides known to be active against a specific strain, the model targets the *joint distribution* of properties unique to that species, acting as a proxy for species-specific constraints without requiring explicit, data-scarce strain labels.
> >
> > Regarding the inclusion of species labels or motifs:
> > *   **Species Labels:** We avoided explicit species conditioning (binary 1/0 tags) as it often requires training strain-specific classifiers. Given the limited data, these classifiers introduce significant estimation errors (T.1, L270-283) that would propagate noise into the generative process. Please refer to Sec.1 of our reply to CTVM for a more extensive analysis.
> > *   **Motifs & Future Work:** Our framework is flexible; more specialized features (e.g., structural motifs) can be added to the conditioning vector as reliable data becomes available, as we illustrate by including the Instability Index to the conditioning properties in an experiment requested by reviewer CVTM (again in section 1. of our response to this reviewer). Furthermore, our diffusion paradigm and residue-level embedding allow for future integration of motif-based guidance via differentiable loss terms during the reverse diffusion process [1].
> >
> > Ultimately, this demonstrates the flexibility and generalizability of our framework, offering a practical improvement over noise-sensitive label-based models by enabling targeted generation, even in data-scarce settings.
> >
> > &nbsp;
> >
> > ### 5. Validation of Species-Specific Classifiers
> >
> > > species-specific “success” [...] supported by elevated classifier scores [...] “mean net charge” is highly dominant; [...] species specificity merely matching certain global attribute distributions.
> >
> > We address the concern that classifiers just match global attributes:
> > *   **Statistically Significant Discrimination:** T.15 (L1405-1419) demonstrates high AUPRC and exceptional LR+ $>200$. Since our negative training data includes peptides active against other bacteria (which share global attributes like charge), this performance is significant: if the models merely tracked global distribution shifts, they would fail to distinguish these sequences.
> > *   **Experimental Evidence:** T.21 (L1674-1682) corroborates this by showing that different species classifiers yield distinct predictions for the same 25 validated peptides, refuting the argument that they rely on a singular global feature.
> > *   **Expected Biological Correlation:** Antimicrobial activity of AMPs is naturally correlated across species [2], perfect classifiers will share feature reliance.
> >
> > &nbsp;
> >
> > We sincerely thank Reviewer ieQ4 for their suggestions. The additional experiments have enriched our manuscript. We hope that our responses address all concerns satisfactorily and look forward to the discussion phase.
> >
> > ---
> > [1] Dhariwal, P., et al. Diffusion models beat gans on image synthesis. NeurIPS
> >
> > [2] Witten, J., et al. Deep learning regression model for antimicrobial peptide design bioRxiv

---

### Official Review · Reviewer_o3XC · 2025-11-06

**Soundness:** 3
**Presentation:** 3
**Contribution:** 3
**Rating:** 6
**Confidence:** 5

**Summary:**

This paper proposes OmegAMP, a controllable diffusion-based framework for antimicrobial peptide (AMP) sequence design. OmegAMP represents each amino acid using physiochemically-inspired property values to obtain biologically-informed peptide embeddings. It further proposes property conditioning techniques to guide the generation toward desired functional characteristics. In addition, the authors introduce synthetic negative sample augmentation for classifier training to reduce false positives in AMP prediction.

**Strengths:**

- The authors introduce practical conditioning methods that offer fine-grained control over desired AMP properties, leading to state-of-the-art generation performance.
- Comprehensive experiments are presented, covering both in silico and wet-lab validation. The 96% experimental hit rate is particularly impressive.
- The authors theoretically and empirically justify the synthetic negative sampling strategy, alleviating concerns about inadvertently labeling true AMPs as negatives.

**Weaknesses:**

- Could you clarify the motivation for including “is AMP” in the conditioning vector? The proposed model is specialized for AMP generation, so, as stated in the manuscript, this entry is always 1 for all training and inference samples.
- The authors mention that PC offers flexibility but may not capture inherent correlations between properties. However, empirical results show PC often performs better than SC. Could the authors provide more intuition for this?
- The embedding quality should be compared against ESM2 embeddings, which are frequently used in AMP generation baselines.
- The reconstruction loss (Eq. 3) seems to minimize the discrepancy against the target sequence’s embedding. In that case, shouldn’t Eq. 3 use $E$ instead of $s$?
- Minor) Table 11 would be more informative if presented in the main manuscript, since controllability result is central to fully assessing the generation performance.

**Questions:**

- How does the performance change when conditioning on the average values of each numerical property (e.g., charge, hydrophobicity) within a predefined range, instead of randomly sampling the property values?
- How many AMPs are used to construct $\mathcal S_{\text{target}}$ in the SC?

---

> ### Author Response · Authors · 2025-11-21
> **Part 1/2**
>
> We are grateful to reviewer o3XC for their positive assessment and support for our work. We appreciate the recognition of the fine-grained control over desired AMP properties offered by our model, and the acknowledgment of the state-of-the-art performance that our methods exhibit in key stages of the AMP discovery pipeline.
>
> &nbsp;
>
> ### 1. Motivation for the "isAMP" Conditioning Vector
>
> > Could you clarify the motivation for including “is AMP” in the conditioning vector? [...] as stated in the manuscript, this entry is always 1 for all training and inference samples.
>
>
> We thank the reviewer for their question. Importantly, the “isAMP” property is set to 0 for all general peptide sequences that we include at the training stage. We have updated the manuscript to better clarify the following aspects (see L160-163):
>
> *   **Inference:** “isAMP” is always set to 1.
> *   **Training (Curated):** “isAMP” is set to 1 if the sequence is in “AMP Sequences” (originating from curated databases).
> *   **Training (Predicted):** “isAMP” is set to 0 if the sequence is in “General Peptide Sequences” (predicted AMPs from Peptipedia). See Appendix C for more dataset details (L877-943).
>
>
> Inspired by your question and reviewer ieQ4, see section 3 of our response to this reviewer, we have validated the inclusion of “General Peptide Sequences” empirically in an ablation (see Appendix I.1 L1238-1259).
>
> **Motivation**
> *   **Data Diversity:** By using such a large dataset, we expose the model to a much larger and chemically diverse set of sequences during training, improving overall generation quality and controllability.
> *   **Learned Correlations:** The rationale is that the connection between physicochemical properties and amino acid sequence can also be learned from general peptides.
> *   **Source Distinction:** We preserve the distinction between these two data sources to enable the generative model to isolate critical patterns related to verified antimicrobial activity.
>
>
> &nbsp;
>
> ### 2. Performance Analysis: Property Conditioning (PC) vs. Sequence Conditioning (SC)
>
> > The authors mention that PC offers flexibility but may not capture inherent correlations between properties. However, empirical results show PC often performs better than SC. Could the authors provide more intuition for this?
>
> We thank the reviewer for this insightful question.
> We explore the disconnect between theoretical expectations and empirical observations below:
>
> *   **Theoretical Expectation:** SC should show superior performance as it samples from the property distributions of known active peptides, thereby respecting natural correlations. As opposed to PC, which samples each property independently.
> *   **Practical Reality:** As shown by the requested experiment regarding conditioning on specific property pairs (details in section 6. of this response), the generative model performs non-uniformly across the property space.
>
> Therefore, we attribute the empirical performance of PC to the explicit guidance provided by expert knowledge, as these ranges explicitly guide the model toward "high-hit-rate" zones, i.e. regions where the model is most robust, whereas SC may sample from valid but harder-to-model regions.
>
> To validate this conclusion, we compare PC and SC within the same property ranges:
> *  **Experimental Setup:** Instead of the more conservative expert defined ranges - Length: [10,30], Charge: [2,10], Hydrophobicity: [-0.5, 0.8], we leverage the property ranges of the EV AMP set used in SC  - Length: [2,98], Charge: [-4,31], Hydrophobicity: [-2.5:1.23]
> *   **Sampling:** We sample 50k sequences using PC on these expanded property ranges.
>
> We observe that:
> *   **Performance Drop:** When conditioning PC on the property ranges used for SC, SC performs significantly better.
> *   **Confirmation of Theory:** These results align with our analysis: increasing PC ranges to include deviation from "high-hit-rate" zones exacerbates the non-natural aspects associated with independent property sampling.
>
>
> | Gen. Model | HydrAMP-MIC | OmegAMP Class. | Fitness Score | Diversity | Uniqueness | Novelty |
> | :--- | ---: | ---: | ---: | ---: | ---: | ---: |
> | OmegAMP | 33.8 | 10.5 | 0.13 | **0.64** | 94 | 98 |
> | OmegAMP-PC on SC property range | 40.6 | 3.2 | 0.12 | 0.57 | 93 | **100** |
> | OmegAMP-PC | **70.2** | 14.8 | **0.16** | 0.60 | **98** | 99 |
> | OmegAMP-SC | 64.1 | **16.4** | 0.15 | 0.61 | 95 | 97 |
>
> Furthermore, we expect this trend to continue for the comparison between PC and SC in the species specific setting, as the property ranges for these subsets display similar ranges, for instance E. Coli’s ranges are Length: [2, 75], Charge: [-4, 31], Hydrophobicity: [-2.5, 0.74].
>
> We updated section I.5 of the Appendix (L1372-1390) with these insights.
>
> &nbsp;

---

> > ### Author Response · Authors · 2025-11-21
> > **Part 2/2**
> >
> > ### 3. Benchmarking Embedding Quality against ESM2
> >
> > > The embedding quality should be compared against ESM2 embeddings, which are frequently used in AMP generation baselines.
> >
> > We agree with the reviewer regarding the importance of comparing with ESM2 embeddings. As it stands, our experimental setting already addresses this by comparing our model with AMP-Diffusion, which follows a similar latent diffusion approach but operates on the ESM2 representation.
> >
> > To further compare the embedding quality of OmegAMP and ESM2, we considered the following:
> > *   **Embeddings:** We extracted OmegAMP and ESM2 embeddings from the “AMP sequences” of the Generative Dataset (Appendix C in L877-943)
> > *   **Regressor Training:** We trained regressors to predict key properties (charge, hydrophobicity, instability index, Boman index, aliphatic index).
> > *   **Evaluation Strategy:** We performed 5-fold cross-validation and reported the mean absolute error (MAE) for each embedding type.
> >
> > The results suggest that OmegAMP offers an embedding space, where the computation of these properties is much more straightforward, as evidenced by smaller absolute errors.
> >
> > |**Embedding Type**|**Charge**|**Hydrophobicity**|**Instability Index**|**Boman Index**|**Aliphatic Index**|
> > |---|---|---|---|---|---|
> > |ESM-2 (PCA)|0.683 ± 0.007|0.084 ± 0.001|19.55 ± 0.17|0.448 ± 0.005|12.06 ± 0.09|
> > |ESM-2 (Avg. Pooling)|0.556 ± 0.004|0.067 ± 0.000|18.30 ± 0.16|0.359 ± 0.005|10.20 ± 0.08|
> > |OmegAMP|**0.526 ± 0.010**|**0.063 ± 0.001**|**15.70 ± 0.21**|**0.328 ± 0.004**|**8.12 ± 0.15**|
> >
> > We updated section I.2 of the Appendix (L1260-1282) with these insights.
> >
> > Moreover, to highlight the embedding differences, we note that our biologically-inspired embedding is much more compact than ESM2. For reference, the representation dimensions are:
> > *   **OmegAMP embedding:** (6, 100)
> > *   **Smallest ESM2 embedding:** (320, 100)
> > *   **Latent representation:** (32, 25)
> >
> >
> > Consequently, replacing our embedding with ESM2 is not straightforward as it requires a different ( and likely much larger) denoising model, since the latent dimensions must exceed the input dimensions. Such architectural changes would make a direct comparison infeasible.
> >
> > &nbsp;
> >
> > ### 4. Notation Correction
> >
> > > [...] shouldn’t Eq. 3 use $E$ instead of $s$
> >
> > Thank you for highlighting this imprecision. We updated L177-180 of the paper accordingly.
> >
> >
> > &nbsp;
> >
> > ### 5. Manuscript Structure Updates
> >
> > > Table 11 would be more informative if presented in the main manuscript [...]
> >
> > Thank you for the suggestion. We moved the I.3 section of the Appendix to the 4.3 section of the main paper (now in L452-468).
> >
> > &nbsp;
> >
> > ### 6. Controllability on Numerical Property Ranges
> >
> > > How does the performance change when conditioning on [...] numerical properties (e.g., charge, hydrophobicity) within a predefined range [...]
> >
> > Thank you for the suggestion. For completeness, we note that section 4.3 (L452-468) explores an adjacent question, showing that OmegAMP is able to consistently generate sequences within predefined ranges, unlike other models.
> >
> > To fully address the reviewers' concerns, we provide an additional experiment:
> > *   **Analysis Protocol:** We analyze the mean absolute error between conditioned and obtained charge-hydrophobicity pairs (without length conditioning).
> > *   **Sampling Grid:** We form the Cartesian product between charge (0, 2, 4, 6, 8, 10) and hydrophobicity (-0.5, -0.2, 0.0, 0.2, 0.4, 0.6, 0.8) and sample 250 sequences for each pair.
> >
> >
> > The results show:
> > *   **Excellent Controllability:** OmegAMP consistently displays low mean absolute error across charge-hydrophobicity pairs.
> > *   **Biochemical Constraints:** Certain physicochemical regions are more challenging than others; this is in accordance with physicochemical constraints, where increasing charge and hydrophobicity simultaneously becomes biochemically prohibited.
> >
> > The figure is shown in Appendix I.4 (L1313-1369)
> >
> > &nbsp;
> >
> > ### 7. Dataset Composition for Sequence Conditioning
> >
> > > How many AMPs are used to construct $\mathcal{S}_{\text{target}}$ in the SC?
> >
> > There are 4209 sequences used to construct $\mathcal{S}_{\text{target}}$ in SC. For more details, check Appendix C; these sequences are the “General EV”. We updated the paper to provide this information more clearly (L355-357).
> >
> > &nbsp;
> >
> > We once again thank Reviewer o3XC for their constructive feedback, which has prompted meaningful additional experiments, particularly regarding the embedding comparison and conditioning strategies, that have significantly enriched our manuscript. We look forward to engaging in further discussion.

---

### Author Response · Authors · 2025-12-01
**Summary of Strengths & Rebuttal Outcomes**

Dear Area Chair,

We sincerely appreciate your time evaluating our submission. Given the unusual rebuttal constraints posed by the ICLR 2026 data leak, we provide a concise summary of our paper’s strengths and the rebuttal outcomes.

### Summary of Strengths

* **Versatile Conditioning Mechanism:** We introduced a flexible conditioning architecture that injects physicochemical constraints (e.g., charge, hydrophobicity) directly into the diffusion process. By leveraging our novel **Property and Subset Conditioning strategies**, we enabled the targeted generation of peptides active against specific bacterial strains, providing “a novel and practical way to achieve fine-grained and species-targeted controllability” (Reviewer CVTM).
* **Biologically Informed Embeddings:** We replaced standard encodings with a domain-specific embedding scheme derived from five physicochemical scales. This inductive bias proved critical for modeling antimicrobial function, allowing OmegAMP to outperform existing models and achieve “state-of-the-art generation performance” (Reviewer o3XC).
* **Robust Filtering (<1% FPR):** Our novel augmentation strategy and weighted loss function produced “low–false-positive filtering classifiers” (Reviewer ieQ4). By reducing misclassification rates to <1% on challenging benchmarks (e.g., natural non-AMPs like signal and metabolic peptides), we directly address “the chronic high-FPR problem in AMP screening” (Reviewer CVTM).
* **Unprecedented Experimental Success:** As highlighted by reviewer o3XC, OmegAMP achieves a “particularly impressive” wet-lab success rate of 96% (24 of 25 candidates showed antimicrobial activity $\leq 32$ µg/ml against at least one bacterial strain). This significantly outperforms baselines like AMP-Diffusion (\~73%) and HydrAMP (\~58%).
* **Clinical Relevance (MDR Efficacy):** Our peptides demonstrated a 92% success rate against **multi-drug resistant (MDR)** pathogens at low concentrations (8 µg/ml). Reviewer CVTM noted that this efficacy signals a “high potential impact for antimicrobial discovery.”

---

### Summary of Rebuttal Outcomes

During the rebuttal phase, we thoroughly addressed all reviewers’ concerns with substantial new experiments and analyses.

**1. Validated Design Choices & Robustness**
  * We confirmed the usefulness of our embeddings through ablations, showing our biologically informed embeddings **outperform ESM-2 in property prediction**.
  * We performed a sensitivity analysis demonstrating that the classifier maintains high performance even when **\~30% of positive training labels are corrupted**, reinforcing the soundness of our weighted loss function.
  * Ablations on **training data composition** confirmed that diverse training data is critical for controllability, reducing physicochemical errors (MAE) from **0.27 to 0.16–0.18** while improving biological plausibility.
  * Further experiments on **conditioning strategies** clarified that while Property Conditioning enforces **expert-defined ranges**, Subset Conditioning is better positioned to capture the **implicit correlations** of specific bacterial targets.

**2. Demonstrated Generalizability and Flexibility**
  * We showcased the framework's extensibility by successfully retraining OmegAMP to **control a new "Instability Index" constraint**.
  * We verified **precise property controllability across a broad grid** of physicochemical ranges.
  * We further highlighted the flexibility of conditioning on **deterministically computable features rather than noisy predicted binary labels**.

**3. External Verification**
  * We validated our superior generative performance using **independent, third-party classifiers (amPEPpy and AMPlify)**, achieving **~84-87% predicted activity** to confirm biological potential beyond internal metrics.

We believe our extensive in-silico and wet-lab validation, strengthened by the rebuttal updates, **establishes OmegAMP as a significant advance in combating antimicrobial resistance.**

---

### Manuscript Updates

|Topic / Update Summary|In Response To|Location in Manuscript|
|:---|:---|:---|
|**Embeddings vs. ESM-2:** Validated superiority of biological embeddings|o3XC|**App. I.2** (L1260-1282)|
|**Label Noise Sensitivity:** Robustness against ~30% label corruption|ieQ4|**App. J.3** (L1469-1499)|
|**Data Composition:** Diversity improves MAE (0.27 $\to$ 0.16) & plausibility|ieQ4|**App. I.1** (L1236-1258)|
|**Conditioning Logic:** Property (ranges) vs. Subset (implicit correlations)|o3XC|**App. I.5** (L1372-1390)|
|**New Constraint:** Retraining for "Instability Index" control|CVTM & ieQ4|**App. M** (L1728-1746)|
|**Controllability Grid:** Verified precision across physicochemical ranges|o3XC|**App. I.4** (L1331-1369)|
|**Deterministic Features:** Clarified use of computable features vs. predicted labels|CVTM|**Main Text** (L162-164)|
|**External Validation:** Validated via amPEPpy and AMPlify (~84-87%)|ieQ4|**App. M** (L1748-1768)|

---

### Meta-Review · Area_Chair_Wf1V · 2026-01-06

**Summary:**

The paper introduces OmegAMP, a comprehensive framework designed to address the high false-positive rates and limited controllability prevalent in Antimicrobial Peptide (AMP) discovery. The system combines a diffusion-based generative model using explicit "biologically informed" embeddings (e.g., charge, hydrophobicity) with an XGBoost classifier trained on "hard negatives" (shuffled and mutated sequences). The study culminates in a wet-lab validation demonstrating a 96% hit rate (24/25) against Multi-Drug Resistant (MDR) pathogens.

Strengths:

(1) Introduce practical conditioning methods that offer fine-grained control over desired AMP properties, leading to state-of-the-art generation performance.

(2) The paper is well-structured with precise methodological detail (conditioning objective, CADS, embedding injectivity/decoding, classifier features/loss), clear figures/tables, and actionable guidance for practitioners.

(3) Comprehensive experiments are presented, covering both in silico and wet-lab validation. The 96% experimental hit rate is particularly impressive.

Weaknesses:

(1) The proposed OmegAMP framework does not introduce new theoretical paradigms or foundational model architectures (e.g., novel backbone networks). Its core generative component is a conditioned diffusion model, and the conditioning relies on explicit, biologically informed embeddings (e.g., charge, hydrophobicity). This approach is perceived more as effective feature engineering rather than a substantial methodological innovation tailored for AMPs. The framework lacks specialized modules or an advanced workflow that represents a clear conceptual advance over existing conditional generation models in this domain.

(2) The embedding quality should be compared against ESM2 embeddings, which are frequently used in AMP generation baselines. Scoring generated samples with the authors’ own classifier inevitably introduces system-internal consistency bias (despite providing HydrAMP-MIC as a secondary scorer).

(3) The current cond(·) vector captures AMP label, length, charge, and hydrophobicity, but omits other relevant biophysical/ADMET factors (e.g., protease stability, hemolysis, toxicity, aggregation, secondary structure motifs). Extending the conditioning space and validating multi-objective trade-offs would increase practical utility.

(4) Several important references are needed.

[1] Artificial intelligence using a latent diffusion modelenables the generation of diverse and potentantimicrobial peptides.

[2] EBAMP: An efficient de novo broad-spectrum antimicrobial peptide discovery framework.

**Reviewer Concerns:**

Reviewers raised several concerns, including the "lack of comparison with ESM2 embeddings" and the risk that "scoring generated samples with the authors’ own classifier inevitably introduces system-internal consistency bias". Additionally, concerns were raised regarding "dataset and bias considerations," specifically that the use of general peptides from Peptipedia and labels from DBAASP might introduce selection bias and label noise, requiring more analysis on data quality and sensitivity to dataset composition.

To address these concerns, the authos added comparative experiments against "ESM-2 (PCA)" and "ESM-2 (Avg. Pooling)"; introduced "more external independent classifiers amPEPpy and AMPlify" for evaluation to mitigate bias; and also "re-trained" the model adding the "Instability Index". Furthermore, the authors have provided constructive responses, including those concerning the novelty of the paper, as well as the clear quantification of how such noise affects the generative distribution and controllability.

However, after I reading this paper carefully myself, two main issues remain: (1) The OmegAMP framework does not introduce new theoretical paradigms or foundational architectures, such as a novel backbone network. (2) Although the wet-lab validation with a 96% hit rate is a strong point, it is perceived partly as offsetting the methodological limitations. The experimental evidence, while impressive in outcome, lacks depth in methodological detail (e.g., synthesis protocols) and could benefit from more diverse analytical experiments to robustly substantiate the framework's effectiveness beyond the final success rate.

**Reviewer Scores:**

The paper received mixed initial ratings: one borderline accept and two borderline rejects (scores of 6, 4, and 4). While some scores could be shifted during discussion, it remains a borderline paper overall. As noted, its core strength lies in being a solid, complete applied bio-discovery workflow. However, its machine learning components are perceived as lacking the novelty and theoretical depth typically expected for this venue.

---

### Decision · Program_Chairs · 2026-01-26

Reject